# A feed-forward pathway drives LRRK2 kinase membrane recruitment and activation

Edmundo G Vides[1,2], Ayan Adhikari[1,2], Claire Y Chiang[1,2], Pawel Lis[2,3], Elena Purlyte[3], Charles Limouse[1], Justin L Shumate[1], Elena Spínola-Lasso[3,4], Herschel S Dhekne[1,2], Dario R Alessi[2,3], Suzanne R Pfeffer[1,2]*

[1]Department of Biochemistry, Stanford University, Stanford, United States; [2]Aligning Science Across Parkinson's (ASAP) Collaborative Research Network, Chevy Chase, United States; [3]MRC Protein Phosphorylation and Ubiquitylation Unit, University of Dundee, Dundee, United Kingdom; [4]Instituto Universitario de Investigaciones Biomédicas y Sanitarias (IUIBS), Departamento de Bioquímica y Biología Molecular, Universidad de Las Palmas de Gran Canaria, Gran Canaria, Spain

*For correspondence:
pfeffer@stanford.edu

**Abstract** Activating mutations in the leucine-rich repeat kinase 2 (LRRK2) cause Parkinson's disease, and previously we showed that activated LRRK2 phosphorylates a subset of Rab GTPases (Steger et al., 2017). Moreover, Golgi-associated Rab29 can recruit LRRK2 to the surface of the Golgi and activate it there for both auto- and Rab substrate phosphorylation. Here, we define the precise Rab29 binding region of the LRRK2 Armadillo domain between residues 360–450 and show that this domain, termed 'site #1,' can also bind additional LRRK2 substrates, Rab8A and Rab10. Moreover, we identify a distinct, N-terminal, higher-affinity interaction interface between LRRK2 phosphorylated Rab8 and Rab10 termed 'site #2' that can retain LRRK2 on membranes in cells to catalyze multiple, subsequent phosphorylation events. Kinase inhibitor washout experiments demonstrate that rapid recovery of kinase activity in cells depends on the ability of LRRK2 to associate with phosphorylated Rab proteins, and phosphorylated Rab8A stimulates LRRK2 phosphorylation of Rab10 in vitro. Reconstitution of purified LRRK2 recruitment onto planar lipid bilayers decorated with Rab10 protein demonstrates cooperative association of only active LRRK2 with phospho-Rab10-containing membrane surfaces. These experiments reveal a feed-forward pathway that provides spatial control and membrane activation of LRRK2 kinase activity.

## Editor's evaluation

This article, which is of interest to membrane biologists and colleagues in signal transduction, examines the interesting question of whether LRRK2 recruitment to membranes may regulate its activity. Membrane association involves binding to membrane-tethered Rab GTPases via LRRK2's Armadillo domain, and the authors provide an exciting and elegant feed-forward mechanism to describe how recruitment of phospho-RAB8 can promote phosphorylation of RAB10.

## Introduction

Activating mutations in the leucine-rich repeat kinase 2 (LRRK2) cause inherited Parkinson's disease and lead to the phosphorylation of a subset of Rab GTPases (*Alessi and Sammler, 2018*; *Pfeffer, 2022*), in particular, Rab8A, Rab10, and Rab29 within a conserved residue of the Switch II effector-binding motif. Rab GTPases are master regulators of membrane trafficking and are thought to serve

as identity determinants of membrane-bound compartments of the secretory and endocytic pathways (*Pfeffer, 2017*). In their GTP-bound forms, Rabs are best known for their roles in linking motor proteins to transport vesicles and facilitating the process of transport vesicle docking.

Our previous work showed that Rab phosphorylation blocks the ability of Rab proteins to be activated by their cognate guanine nucleotide exchange factors or to bind to the GDI proteins that recycle GDP-bearing Rabs from target membranes to their membranes of origin (*Steger et al., 2016*; *Steger et al., 2017*). Moreover, phosphorylation of Rab8A and Rab10 blocks their ability to bind known effector proteins and enhances binding to a novel set of effectors that includes RILPL1, RILPL2, JIP3, JIP4, and MyoVa proteins (*Steger et al., 2017*; *Waschbüsch et al., 2020*; *Dhekne et al., 2021*). Thus, Rab phosphorylation flips a switch on Rab effector selectivity that can drive dominant physiological changes, including blocking primary cilia formation (*Steger et al., 2017*; *Dhekne et al., 2018*; *Sobu et al., 2021*; *Khan et al., 2021*) and autophagosome motility in axons (*Boecker et al., 2021*).

Most LRRK2 is found in the cell cytosol where it appears to be inactive (*Biskup et al., 2006*; *Berger et al., 2010*; *Purlyte et al., 2018*). Recent structural analysis of the catalytic, C-terminal half of LRRK2 (*Deniston et al., 2020*) and full-length human LRRK2 protein yielded structures of both monomeric and dimeric, inactive states (*Myasnikov et al., 2021*). Several groups have reported that active LRRK2 is a dimer (*Greggio et al., 2008*; *Klein et al., 2009*; *Sen et al., 2009*; *Berger et al., 2010*; *Civiero et al., 2012*; *Guaitoli et al., 2016*), and higher-order forms were detected on membranes upon cross-linking (*Berger et al., 2010*; *Schapansky et al., 2014*) and upon Rab29 binding (*Zhu et al., 2022*). Thus, LRRK2 membrane association is associated with kinase activation; however, the molecular basis for this activation is not yet known.

Exogenously expressed, Golgi-localized Rab29 protein can recruit LRRK2 onto membranes and activate it there for both auto- and Rab substrate phosphorylation (*Kuwahara et al., 2016*; *Liu et al., 2018*; *Purlyte et al., 2018*; *Madero-Pérez et al., 2018*). Indeed, even Rab29 artificially anchored on mitochondria can activate LRRK2 and drive its membrane recruitment (*Gomez et al., 2019*). *McGrath et al., 2021* implicated LRRK2 residues 386–392 as being important for the interaction of Rab29/32/38 family members with the LRRK2 kinase Armadillo domain. However, the LRRK2 Armadillo domain is located at some distance from the kinase domain, at least in the current structure models for LRRK2 protein (*Myasnikov et al., 2021*). Thus, how Rab29 binding might activate LRRK2 kinase activity is not at all clear. In addition, because Rab29 is not needed for LRRK2 action on Rab8A or Rab10 proteins (*Kalogeropulou et al., 2020*), other pathways for LRRK2 activation must exist.

In this study, we define a specific patch ('site #1') of the LRRK2 Armadillo domain that binds to Rab8A, Rab10, and Rab29 protein with affinities similar to those reported previously (*McGrath et al., 2021*). More importantly, we identify a distinct region of LRRK2 Armadillo domain ('site #2') that binds specifically to LRRK2-*phosphorylated* Rab8A and Rab10 proteins, to establish a feed-forward activation mechanism for membrane-associated LRRK2 kinase.

## Results

### Rab29 binds to the C-terminal portion of the LRRK2 Armadillo domain

*McGrath et al., 2021* showed that the LRRK2 Armadillo domain residues 1–552 contain a binding site that interacts specifically with purified Rab29, 32, and 38 in vitro with affinities of 2.7, 1.2, and 1.2–2.4 µM, respectively. We used microscale thermophoresis to determine the affinity of other Rab GTPase substrates with this portion of LRRK2 kinase. For these experiments, portions of the LRRK2 Armadillo domain were fluorescently labeled and incubated with Rab GTPases in the presence of $Mg^{2+}$-GTP. *Figure 1* shows binding curves for Rab29 with full-length Armadillo domain (residues 1–552, *Figure 1A*), as well as sub-fragments composed of LRRK2 residues 1–159 (*Figure 1C*) or 350–550 (*Figure 1D*). Rab29 showed specific binding to the full-length 1–552 Armadillo fragment with a $K_D$ of 1.6 µM (*Figure 1A*), comparable to that reported previously using other methods (*McGrath et al., 2021*). Under these conditions, the non-LRRK2 substrate Rab7 protein failed to bind to the Armadillo 1–552 fragment (*Figure 1B*). No Rab29 binding was detected to a fragment representing the N-terminal 1–159 LRRK2 residues (binding >29 µM; *Figure 1C*); essentially full binding was observed with a fragment encompassing residues 350–550 ($K_D$ = 1.6 µM; *Figure 1D*). Thus, Rab29 binds to the C-terminal portion of LRRK2's Armadillo domain at a site that we will refer to as site #1.

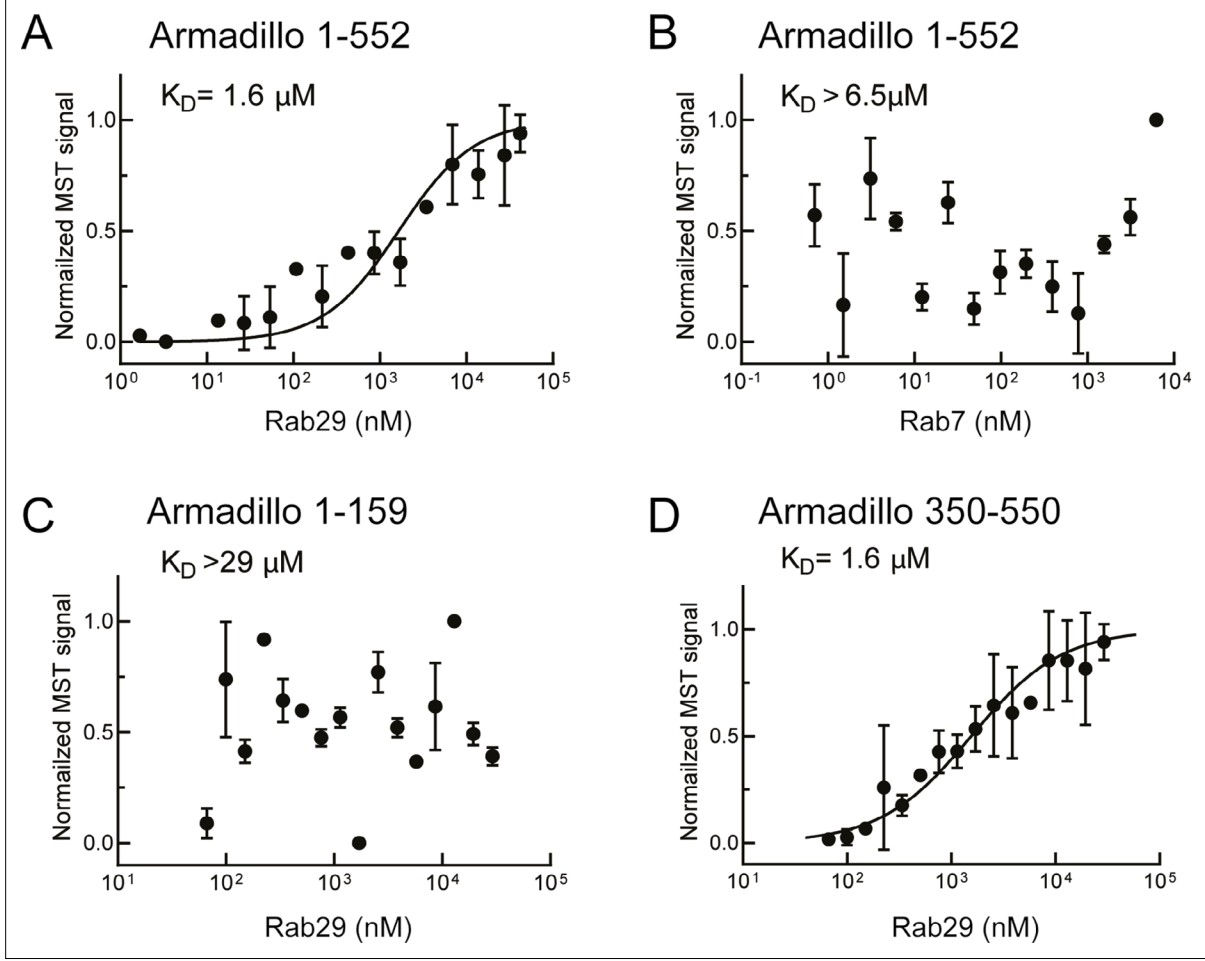

**Figure 1.** Rab29 binds to the C-terminal portion of the LRRK2 Armadillo domain. Microscale thermophoresis of full-length (residues 1–552), labeled LRRK2 Armadillo domain with His-Rab29 (**A**) or with His-Rab7 (**B**). (**C, D**) Microscale thermophoresis of labeled LRRK2 Armadillo domain residues 1–159 (**C**) or 350–550 (**D**) with Rab29. Purified Rab29 was serially diluted and then NHS-RED-labeled-LRRK2 Armadillo (final concentration 100 nM) was added. Graphs show mean and SEM from three independent measurements, each from a different set of protein preparations. Data are summarized in **Table 1**.

## Rab8A and Rab10 bind to the LRRK2 Armadillo domain

Similar experiments were carried out with Rab8A and Rab10, the most prominent LRRK2 substrates (**Steger et al., 2017**). Rab8A-bound full-length Armadillo domain with a $K_D$ of 2.9 µM (**Figure 2A**) showed weaker interaction with the LRRK2 1–159 fragment ($K_D$ ~ 6.7 µM; **Figure 2B**) and good binding to the 350–550 fragment ($K_D$ = 2.3 µM; **Figure 2C**). These data indicate that Rab8A may bind to the same site as Rab29. Like Rab8A, Rab10 bound to full-length Armadillo 1–552 with a $K_D$ of 2.4 µM (**Figure 2D**); weaker binding was detected for 1–159 and 350–550 fragments, yielding $K_D$s of 5.1 µM in both cases (**Figure 2E and F**). Thus, in addition to Rab32, 38 and 29, Rabs 8A and 10 can bind to LRRK2 residues 350–550. Note that Rab32 and Rab38 are not substrates of LRRK2 kinase as they lack a phosphorylatable Ser/Thr residue in the Switch II motif (**Steger et al., 2016**; 2107); they show extremely narrow tissue-specific expression but are related to Rab29 protein.

## Residues critical for Rab GTPase binding to LRRK2 residues 350–550: Site #1

Previous work implicated LRRK2 residues 386–392 in contributing to a Rab29/32/38 binding interface (**McGrath et al., 2021**). We used a microscopy-based assay to identify any portions of the first 1000 residues of L.

 RRK2 that would relocalize to the Golgi upon co-expression with Golgi-localized, HA-Rab29 protein (**Figure 3—figure supplement 1**). Twenty-two constructs were transfected into cells and their

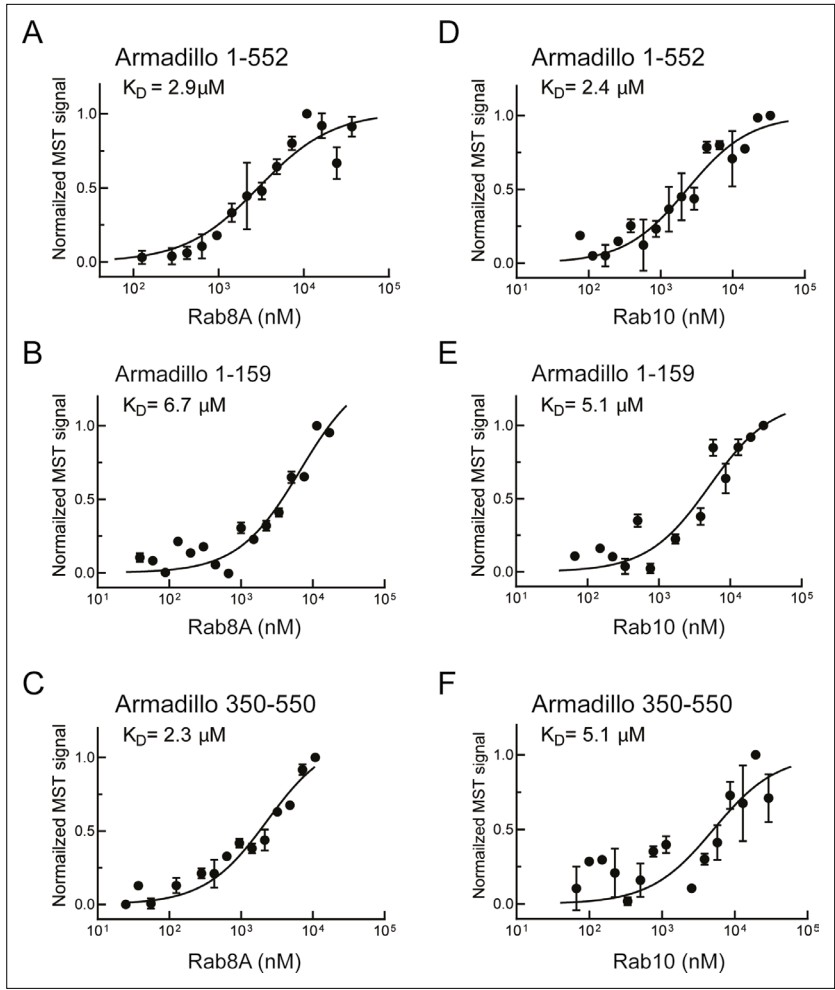

**Figure 2.** Rab8A and Rab10 bind to the LRRK2 Armadillo domain. (**A–C**) Microscale thermophoresis of labeled, LRRK2 Armadillo domain fragments comprised of residues 1–552, 1–159, or 350–550 with Rab8A Q67L as indicated. (**C–E**) Microscale thermophoresis for Rab10 Q68L (1–181) with indicated LRRK2 Armadillo fragments, as in (**A**). Purified Rab proteins were serially diluted and then NHS-RED-labeled LRRK2 Armadillo domain (final concentration 100 nM) was added. Graphs show mean and SEM from three independent measurements, each from a different set of protein preparations. Data are summarized in *Table 1*.

localization scored visually. The smallest fragment of LRRK2 that interacted with HA-tagged Rab29 in HeLa cells, thereby co-localizing at the Golgi complex, encompassed LRRK2 residues 350–550.

We next deployed AlphaFold docking (*Jumper et al., 2021*) using ColabFold (*Mirdita et al., 2022*) and the AlphaFold2_advanced.ipynb notebook with the default settings to model the interaction of Rab29 with the LRRK2 350–550 fragment (*Figure 3A*, *Figure 3—figure supplement 2*). Residues highlighted in red show key contacts between LRRK2 and Rab29 and will be shown below to be essential for detection of this interaction in cells. This modeled structure of site #1 is extremely similar to that of the recently reported experimental cryo-EM structure of Rab29 complexed full-length LRRK2 (*Zhu et al., 2022*).

Three metrics were used to evaluate the importance of individual residues to contribute to Rab29 interaction. First, we tested the impact of mutations on the ability of full-length LRRK2 to co-localize with HA-Rab29 at the Golgi in HeLa cells (*Figure 3B*, *Figure 3—figure supplement 3*); we also tested the ability of exogenously expressed Rab29 to stimulate activity of the same point mutants in the background of either wild-type LRRK2 (*Figure 3C*, *Figure 3—figure supplement 4A*) or pathogenic R1441G LRRK2 (*Figure 3D*, *Figure 3—figure supplement 4B*). This work identified four key mutations of highly conserved residues (R361E, R399E, L403A, and K439E) that blocked both the co-localization

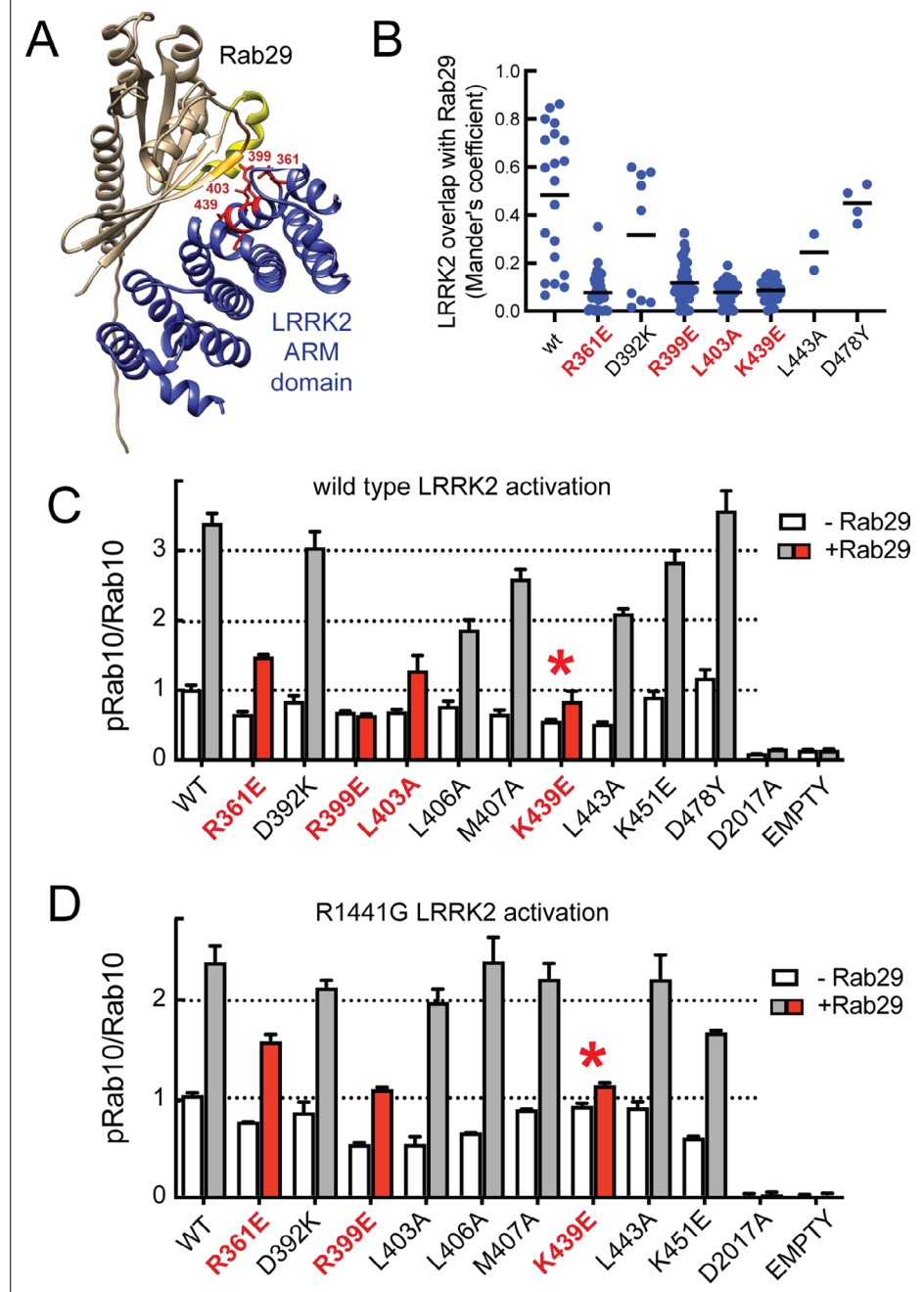

**Figure 3.** Characterization of critical LRRK2 residues mediating binding to Rab29. (**A**). Predicted interactions between Rab29 and the LRRK2 Armadillo domain using AlphaFold docking (***Jumper et al., 2021***), ColabFold (***Mirdita et al., 2022***), and the AlphaFold2_advanced.ipynb notebook default settings. Residues identified in red show key contacts between LRRK2 and Rab29; orange and yellow coloring indicates the Switch I and Switch II domains of Rab29. (**B**) The wild-type and indicated mutants of full length of GFP-LRRK2 were co-expressed with HA-Rab29 in HeLa cells. 24 hr post transfection, cells were fixed and localization assessed by confocal microscopy. LRRK2 overlap with Rab29 is presented as a Mander's coefficient determined using CellProfiler software (***McQuin et al., 2018***). (**C, D**) Wild-type and indicated mutants of full length of GFP-LRRK2 (**C**) or GFP-LRRK2 R1441G (**D**) were co-expressed with HA-Rab29 in HEK293T cells. 24 hr post transfection, cells were lysed and extracts immunoblotted with the indicated antibodies. Shown are the averages and standard deviations of duplicate determinations; red asterisks indicate preferred mutant.

The online version of this article includes the following source data and figure supplement(s) for figure 3:

**Figure supplement 1.** Top: Fragments of GFP-LRRK2 that were co-expressed with HA-Rab29 in HeLa cells.

*Figure 3 continued on next page*

*Figure 3 continued*

**Figure supplement 2.** Residues predicted to be critical for Rab29 interaction with part of the LRRK2 Armadillo domain and comparison with an AlphaFold model for the complex with Rab8A.

**Figure supplement 3.** Examples of micrographs used to create *Figure 3B*.

**Figure supplement 4.** Immunoblots used to obtain *Figure 3C and D*.

**Figure supplement 4—source data 1.** Raw data for gels.

**Figure supplement 4—source data 2.** Annotated gels.

of LRRK2 and Rab29 in HeLa cells (*Figure 3B*, red), as well as activation of LRRK2 upon overexpression of Rab29 in HEK293 cells (*Figure 3C*, red). In experiments undertaken with pathogenic R1441G LRRK2 that is more potently activated by Rab29, the K439E LRRK2 mutation completely blocked LRRK2 kinase activation; R399E showed weak activation (*Figure 3D*, *Figure 3—figure supplement 4B*). Some of the other mutants blocked co-localization with Rab29 in HeLa cells without completely suppressing LRRK2 activation following overexpression of Rab29. We therefore recommend using the site#1 K439E LRRK2 mutation to block Rab29 interaction and activation in future work (asterisks in *Figure 3B and C*) as it shows the lowest amount of Rab29 activation with pathogenic R1441G LRRK2. Altogether, these data highlight the importance of a surface that is comprised of LRRK2 residues Arg361, Arg399, Leu403, Lys439 in binding Rab GTPases (site #1) (*Figure 3A*, *Figure 3—figure supplement 2*). Analysis of Rab8A interaction with the LRRK2 350–550 fragment using AlphaFold within ChimeraX 1.4 confirmed the importance of the same LRRK2 residues for Rab8A interaction in silico (*Figure 3—figure supplement 2C*).

## PhosphoRab binding to LRRK2: Site #2

To understand the consequences of LRRK2-mediated Rab GTPase phosphorylation, it is important to identify specific binding partners of phosphorylated Rab proteins and study the consequences of such binding events. We recently established a facile method that enables us to monitor phosphoRab binding to proteins of interest in conjunction with microscale thermophoresis binding assays. Briefly, Rab proteins are phosphorylated >90% in vitro by MST3 kinase (*Dhekne et al., 2021*; *Vides and Pfeffer, 2021*) that phosphorylates Rab proteins at the same position as LRRK2 kinase (*Vieweg et al., 2020*). Recombinant MST3 is much easier to purify in large amounts for biochemical experiments than LRRK2. We used this assay to monitor the possible interaction of phosphorylated LRRK2 substrates to the LRRK2 Armadillo domain and were delighted to discover that pRab8A and pRab10 proteins bind with high affinity to a site distinct from that used by non-phosphorylated Rab proteins that we term site #2.

As shown in *Figure 4*, phosphoRab8A and phosphoRab10 bound with $K_D$s of ~900 nM and 1 μM to the full Armadillo domain 1–552 fragment, respectively (*Figure 4A and D*); this binding reflected interaction with N-terminal LRRK2 residues 1–159 as this fragment was sufficient to yield essentially the same $K_D$s of 1 μM and 700 nM, respectively, for phosphoRab8A and phosphoRab10 proteins (*Figure 4B and E*). Furthermore, no binding was detected for phosphoRab8A or phosphoRab10 with LRRK2 residues 350–550 (*Figure 4C and F*). These data demonstrate that Rab8A and Rab10 GTPases, phosphorylated at the same residues modified by LRRK2 kinase, bind very tightly to the LRRK2 N-terminus but no longer interact with the 350–550 region that interacts with dephosphorylated Rab proteins.

Note that non-phosphorylated Rab8A and Rab10 also bound to the site #2-containing fragment 1–159 with relatively weak affinities of 5 or 6 μM (*Figure 2B and E*; *Table 1*). Interestingly, AlphaFold in ChimeraX (*Pettersen et al., 2021*) predicts that the 1–159 fragment contains a potential, non-phosphoRab-binding site that is occluded in a longer fragment (1–400), and thus also in full-length LRRK2. Moreover, as discussed below, these $K_D$ values may be higher than the concentrations of these Rab GTPases in cells, thus it seems unlikely that non-phosphoRabs interact with site #2 under normal physiological conditions. We conclude that phosphoRab binding is the predominant interaction between LRRK2 1–159 and Rab GTPases.

Electrostatic analysis (*Jurrus et al., 2018*, *Pettersen et al., 2004*) of a model of the LRRK2 Armadillo domain revealed that the absolute N-terminus of LRRK2 contains a patch of basic amino

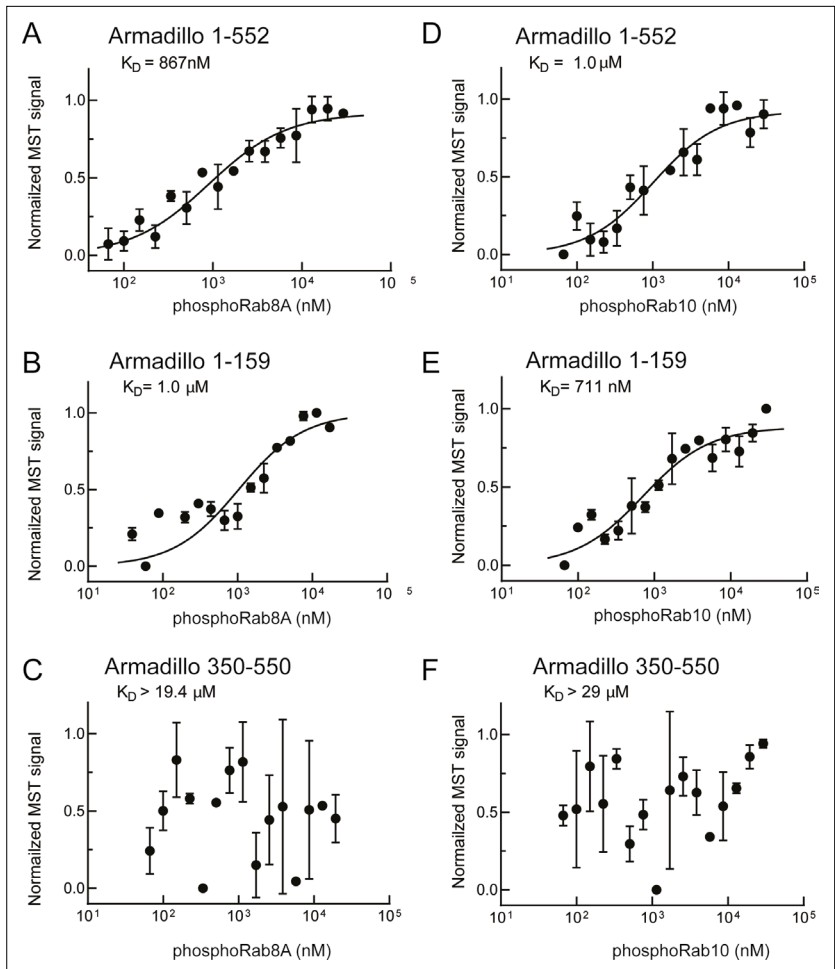

**Figure 4.** PhosphoRab8A and phosphoRab10 bind with high affinity to the N-terminal portion of the LRRK2 Armadillo domain. (**A–F**) Microscale thermophoresis of labeled, indicated, LRRK2 Armadillo fragments with His-phosphoRab8A Q67L (**A–C**) or with His phosphoRab10 Q68L 1–181 (pRab10; **D–F**). Purified Rab proteins were phosphorylated with Mst3 kinase at 27°C for 2 hr and then serially diluted; NHS-RED-labeled Armadillo (final concentration 100 nM) was then added. Graphs show mean and SEM from three independent measurements, each from a different set of protein preparations.

acids (highlighted in blue) that may comprise a phosphoRab interaction interface (*Figure 5A*). Such modeling led us to test the role of lysine residues at positions 17 and 18 in mediating LRRK2 interaction. Mutation of either lysine 17 or 18 abolished phosphoRab10 binding to LRRK2 Armadillo domain, with binding decreased to >20 μM upon single mutation at either site (*Figure 5C and D*). When the conservation score of these residues is analyzed using the Consurf server (*Ashkenazy et al., 2016*), K17 and K18 have a score or 2 and 8, respectively (9 is the maximum score), indicating that K18 is highly conserved and plays an especially important role. These experiments define a second, Rab binding site #2 that is specific for phosphorylated Rab proteins (*Figure 5B*).

To determine the significance of the phosphoRab-binding site in relation to LRRK2 membrane recruitment in cells, we generated full-length FLAG-LRRK2 protein containing point mutations at both lysines 17 and 18 and investigated its cellular localization upon expression in HeLa cells (*Figure 6*). To improve our ability to detect membrane-associated LRRK2 distribution, cells grown on collagen-coated coverslips were dipped in liquid nitrogen and then thawed in a physiological, glutamate-containing buffer to crack open the plasma membrane and release cytosolic proteins prior to fixation (*Seaman, 2004*; *Purlyte et al., 2018*). Under these conditions, LRRK2 co-localizes with phosphorylated Rab proteins (*Purlyte et al., 2018*; *Sobu et al., 2021*).

**Table 1.** Summary of binding affinities.

Note that these values are likely underestimates of affinities as typical preparations of the indicated, purified Rab proteins contained ~50% bound GDP and ~50% bound GTP by mass spectrometry. Non-phosphorylated Rab interaction with Armadillo 1–159 is shown in parentheses as it likely reflects binding to an AlphaFold-predicted site near the C-terminus of this fragment that will not be accessible in full-length LRRK2 protein.

| | Armadillo 1–159 (site #2-containing) | Armadillo 1–552 | Armadillo 350–550 (site #1-containing) | Armadillo 1–552 K17A | Armadillo 1–552 K18A |
|---|---|---|---|---|---|
| Rab29 | >29 | 1.6 ± 0.9 | 1.6 ± 0.5 | - | - |
| Rab10-Q68L | (5.1 ±3.1) | 2.4 ± 0.6 | 5.1 ± 2.5 | - | - |
| pRab10-Q68L | 0.71 ± 0.3 | 1.0 ± 0.4 | >29 | >20 | >20 |
| Rab8A-Q67L | (6.7 ± 3.6) | 2.9 ± 1.2 | 2.3 ± 1.0 | - | - |
| pRab8A-Q67L | 1.0 ± 0.6 | 0.87 ± 0.4 | >19.4 | - | - |
| Rab7 | - | >6.5 | - | - | - |

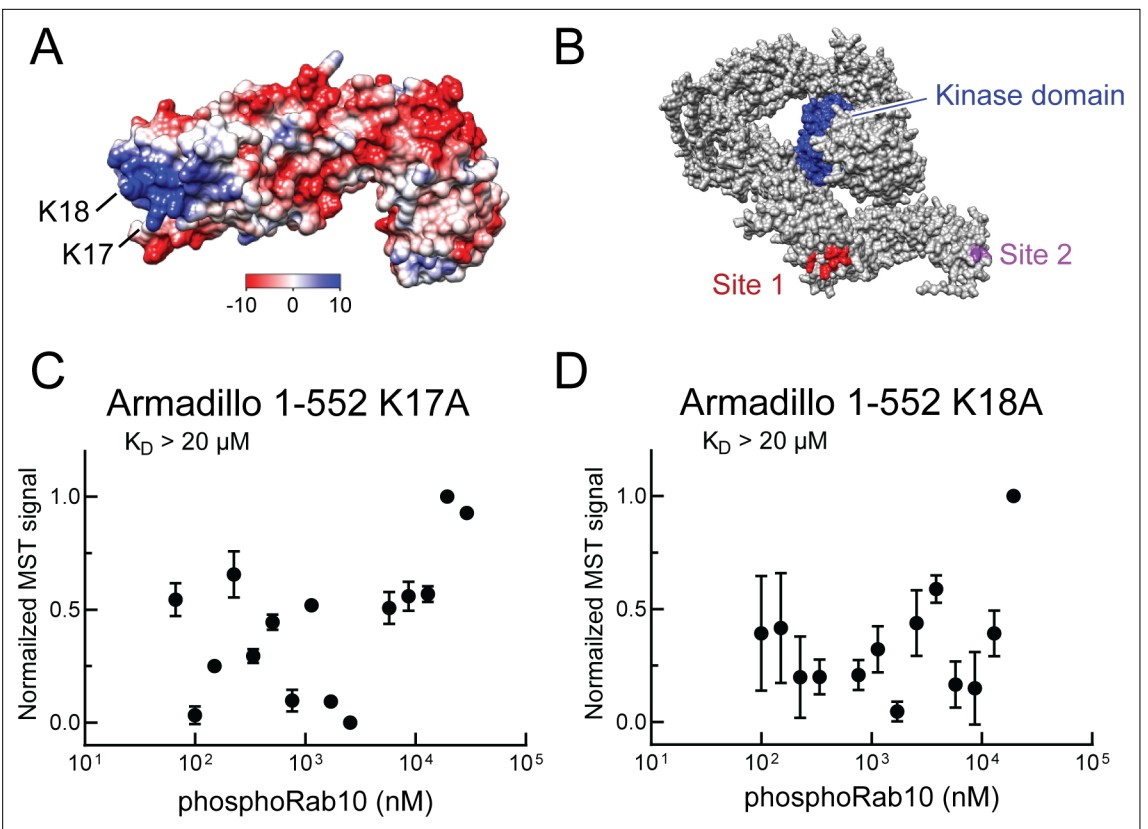

**Figure 5.** Identification of a basic patch at the N-terminus of LRRK2 that is needed for phosphoRab interaction. (**A**) Electrostatic surface potential of LRRK2 Armadillo domain residues 1–552 modeled using Chimera 2 software (***Pettersen et al., 2004***); blue indicates a positively charged surface. LRRK2 K17 and K18 are indicated. (**B**) AlphaFold (***Jumper et al., 2021***) structure of putative, active LRRK2 with residues that mediate Rab29 binding shown in red (site #1) and the K17/K18 residues that are required for phosphoRab10 binding (site #2) shown in magenta; the kinase domain is shown in blue. (**C, D**) Microscale thermophoresis of labeled, full-length LRRK2 K17A or K18A Armadillo 1–552 with His phosphoRab10 Q68L 1–181. Purified Rab10 protein was phosphorylated with Mst3 kinase at 27°C for 2 hr and then serially diluted; NHS-RED-labeled Armadillo (final concentration 100 nM) was then added. Graphs show mean and SEM from three independent measurements, each from a different set of protein preparations.

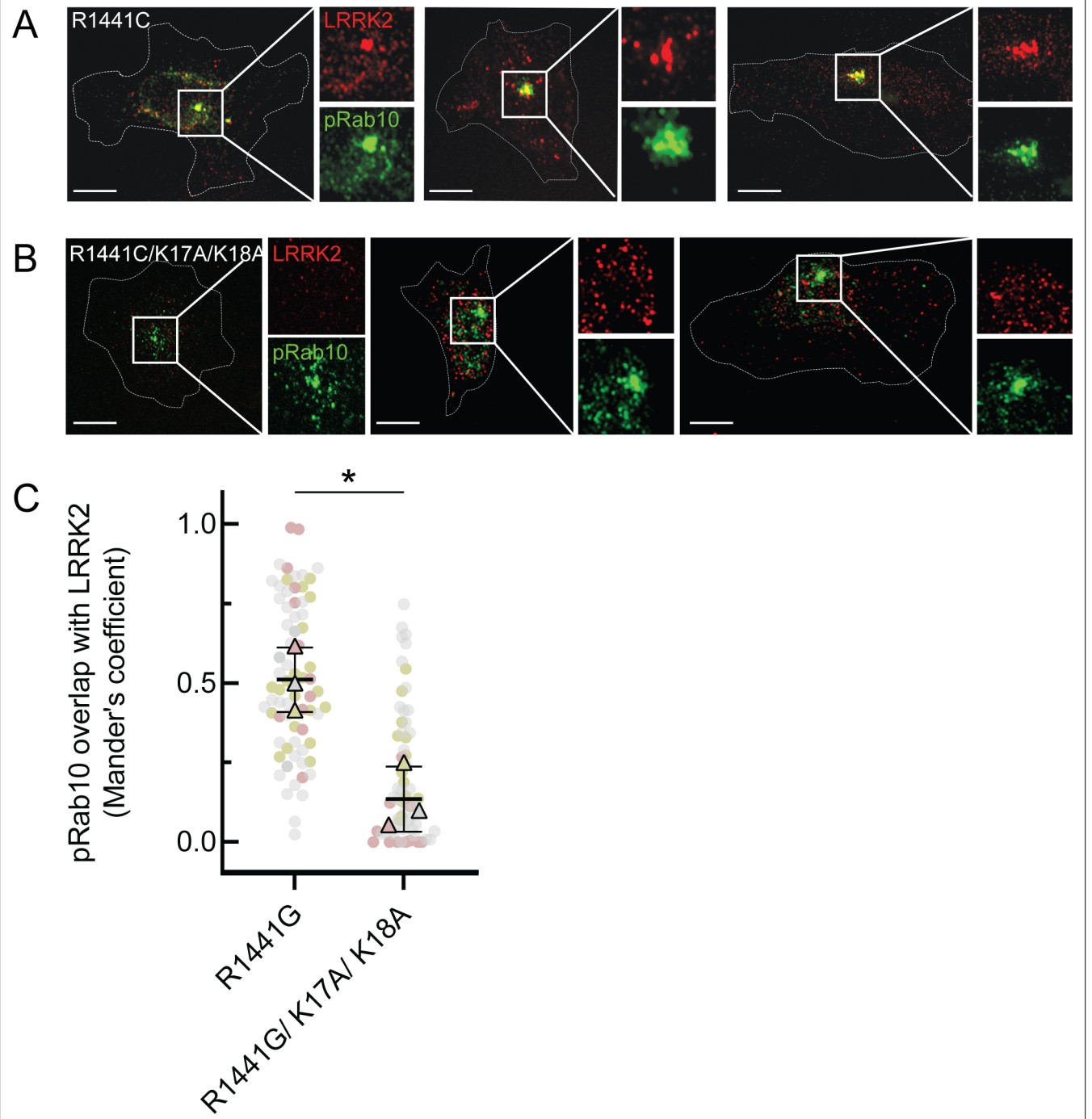

**Figure 6.** LRRK2 K17 and K18 are critical for pRab10 interaction in cells. (**A**) FLAG-LRRK2 R1441G (red) was transfected into HeLa cells plated on collagen-coated coverslips and co-localized with endogenous wild-type pRab10 (green). Cells on coverslips were dipped in liquid nitrogen to deplete cytosol and enhance membrane-bound signal. Insets show enlargements of boxed areas representing peri-centriolar LRRK2 and pRab10. (**B**) FLAG-LRRK2 R1441G/K17A/K18A (red) was transfected into HeLa cells plated on collagen-coated coverslips and stained and localized with pRab10 (green) as in (**A**). Scale bars, 10μm. (**C**) Quantification of pRab10 overlap with LRRK2 by Mander's coefficient. Error bars represent SEM of means from three different experiments (represented by colored dots), each with >40 cells per condition. Significance was determined by *t*-test, *p=0.0108.

As expected, PhosphoRab10 was detected as a bright spot adjacent to the mother centriole in HeLa cells (green, *Figure 6A*), and the co-expressed, R1441G pathogenic mutant LRRK2 protein showed good co-localization with phosphoRab10 protein (red, *Figure 6A and C*), as we have reported previously (*Purlyte et al., 2018*; *Sobu et al., 2021*). In contrast, although exogenously expressed,

R1441G LRRK2 bearing K17/18/A mutations still led to a perinuclear, phosphoRab10-containing structure (green), LRRK2 (red) displayed much less co-localization with the phosphoRab proteins or with membranes overall (*Figure 6B and C*). These experiments show that K17 and K18 are important for exogenous LRRK2 membrane association with a pool of highly phosphorylated Rab10 protein. The importance of LRRK2's N-terminal lysine residues also suggests that caution may be in order when evaluating membrane interactions of LRRK2 tagged N-terminally with larger tags such as GFP that may hinder access to K17/K18.

## PhosphoRab–LRRK2 interaction increases rates of kinase recovery

We next explored the relevance of phosphoRab binding to LRRK2's N-terminus in relation to the overall kinetics of Rab phosphorylation in cells. LRRK2-mediated Rab GTPase phosphorylation is a highly dynamic process that is counteracted by the action of PPM1H phosphatase (*Berndsen et al., 2019*); at steady state, only a small fraction of total Rab proteins are LRRK2-phosphorylated (*Ito*

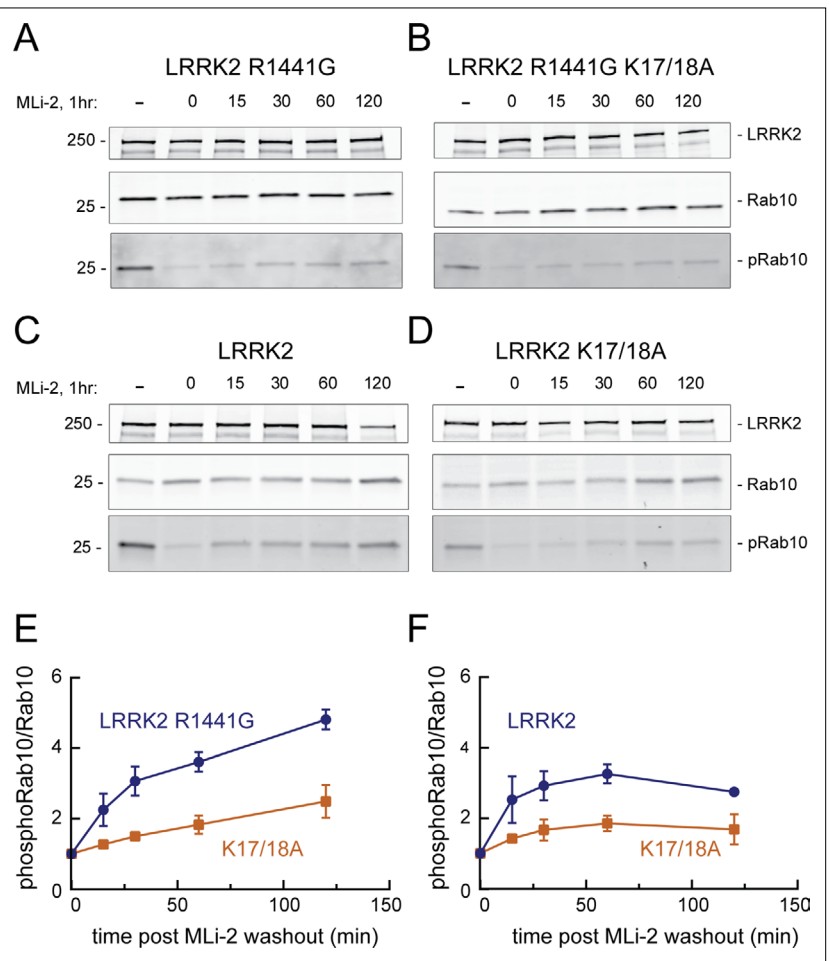

**Figure 7.** LRRK2 K17 and K18 increase endogenous pRab10 recovery after LRRK2 inhibitor washout. (**A–D**) FLAG-LRRK2 R1441G, FLAG-LRRK2 R1441G/K17A/K18A, LRRK2, or LRRK2 K17A/K18A was transfected into HeLa cells. 48 hr post transfection, cells were treated with 200 nM of MLi-2 for 1 hr. The MLi-2 was then removed by multiple washes and incubated for the indicated times prior to cell lysis. Whole-cell extracts (20 μg) were subjected to quantitative immunoblot analysis using anti-LRRK2, anti-Rab10, and anti-pRab10 antibodies. Numbers at the left of the gels represent the mobilities of molecular weight markers in kilodaltons. (**E, F**) Quantification of pRab10/total Rab10 fold change and normalized to no MLi2 control. Error bars represent mean ± SD from two different experiments per condition.

The online version of this article includes the following source data for figure 7:

**Source data 1.** Raw data for gels.

**Source data 2.** Annotated gels.

*et al., 2016*). The initial rate of kinase activity can be determined by monitoring the phosphorylation of Rab10 protein after washout of the LRRK2 inhibitor, MLi-2 (*Ito et al., 2016*; *Kalogeropulou et al., 2020*).

When HeLa cells were treated with 200 nM MLi-2 for 1 hr and then washed with culture medium, Rab10 was efficiently re-phosphorylated by exogenous, FLAG-tagged, R1441G LRRK2 protein over the 2 hr time course evaluated (*Figure 7A and E*). In contrast, cells expressing FLAG-R1441G LRRK2 bearing K17/18A mutations showed comparable total phosphoRab10 levels to begin with, but significantly slower re-phosphorylation (*Figure 7B and E*). Similar results were obtained in experiments comparing the reactivation of FLAG-tagged, wild-type LRRK2 (*Figure 7C and F*) with that of LRRK2 K17/18A (*Figure 7D and F*). As reported previously (*Ito et al., 2016*), wild-type LRRK2 recovery was more efficient than that of R1441G LRRK2. In summary, these experiments demonstrate that K17/K18 residues are important for efficient reactivation of LRRK2 after MLi-2 washout, consistent with their role in anchoring LRRK2 at sites adjacent to phosphorylation substrates.

## Cooperative LRRK2 membrane recruitment on Rab-decorated planar lipid bilayers

Binding of phosphoRabs to site #2 at the N-terminus of LRRK2 (*Figure 5B*) would set up a feed-forward process whereby the product of an initial phosphorylation reaction would enhance subsequent Rab GTPase phosphorylation by holding the enzyme on the surface of membranes that contain relevant Rab GTPase substrates. To visualize the membrane association process directly, we established a planar lipid bilayer system that would enable us to monitor the interaction of fluorescently labeled, purified, full-length LRRK2 kinase with membrane-anchored Rab10 substrate (*Adhikari et al., 2022*). For this purpose, bilayers were formed on the surface of glass-bottom chambers comprised of phospholipids of a composition similar to that found in the Golgi (65% DOPC, 29% DOPS, 1% PI(4)P) (*Thomas and Fromme, 2016*), mixed with 0.01% of the lipophilic tracer DiD dye and 5% DOGS-NTA [Ni$^2$] to enable anchoring of C-terminally His-tagged GFP-Rab10 protein. Binding of fluorescently labeled, hyperactive R1441G LRRK2 was then visualized in real time using total internal reflection (TIRF) light microscopy. Reactions were carried out in the presence of ATP, GTP, and an ATP regenerating system to provide physiological conditions for the full-length LRRK2 enzyme. Note that we routinely utilize R1441G LRRK2 because it is a highly active kinase in cells, although in vitro, R1441G LRRK2 displays the same level of Rab kinase activity as wild-type LRRK2 (cf. *Steger et al., 2017*).

As shown in *Figure 8A* (red dots), fluorescent R1441G LRRK2 bound efficiently to lipid bilayers only in the presence of pre-anchored Rab10 protein (compare with purple dots in 8B) and not when Rab11 protein was instead employed (*Figure 8B*, green dots; *Videos 1–3*). Importantly, almost no binding was observed with kinase inactive D2017A LRRK2 (*Figure 8A*, yellow dots, *Video 4*; *Steger et al., 2016*). This indicates that at least Rab10 GTPase binding to site #1 residues 361–451 results in a low-affinity interaction that is not sufficient to retain this inactive LRRK2 protein on the bilayer under these conditions (7 nM LRRK2, 2.5 μM Rab10). Reactions containing the type I MLi-2 inhibitor showed aggregation of the fluorescent LRRK2 protein, as has been seen in cells. Incubations containing the type 2 inhibitor, GZD-824 (*Tasegian et al., 2021*), showed weak binding, consistent with a requirement for phosphoRab10 generation to support LRRK2 binding to site #2's K17 and K18; however, under these conditions, LRRK2 was not monodisperse and could not be analyzed further. Importantly, R1441G LRRK2 mutated at lysines 17 and 18 bound to a lower extent than R1441G LRRK2 (*Figure 8A*, blue dots; *Video 5*), confirming their important role in binding to phosphorylated Rab8A and Rab10. It is noteworthy that the K17/K18 mutant protein showed higher binding than the D2017A mutant, suggesting that a non-phosphoRab-binding site may be more accessible for binding in an active versus inactive LRRK2 protein conformation.

Analysis of the kinetics of LRRK2 binding as a function of Rab protein concentration showed clear, cooperative membrane association of R1441G LRRK2, consistent with a feed-forward mechanism, as predicted from the in vitro Rab-binding data (*Figure 8C*). A nonlinear regression fit of the data indicated a Hill coefficient of 2.7, consistent with a positive, cooperative phenomenon. In summary, these data demonstrate that LRRK2 kinase is recruited to membranes and then held there by phosphorylated Rabs to increase subsequent Rab GTPase phosphorylation as part of a cooperative, feed-forward pathway.

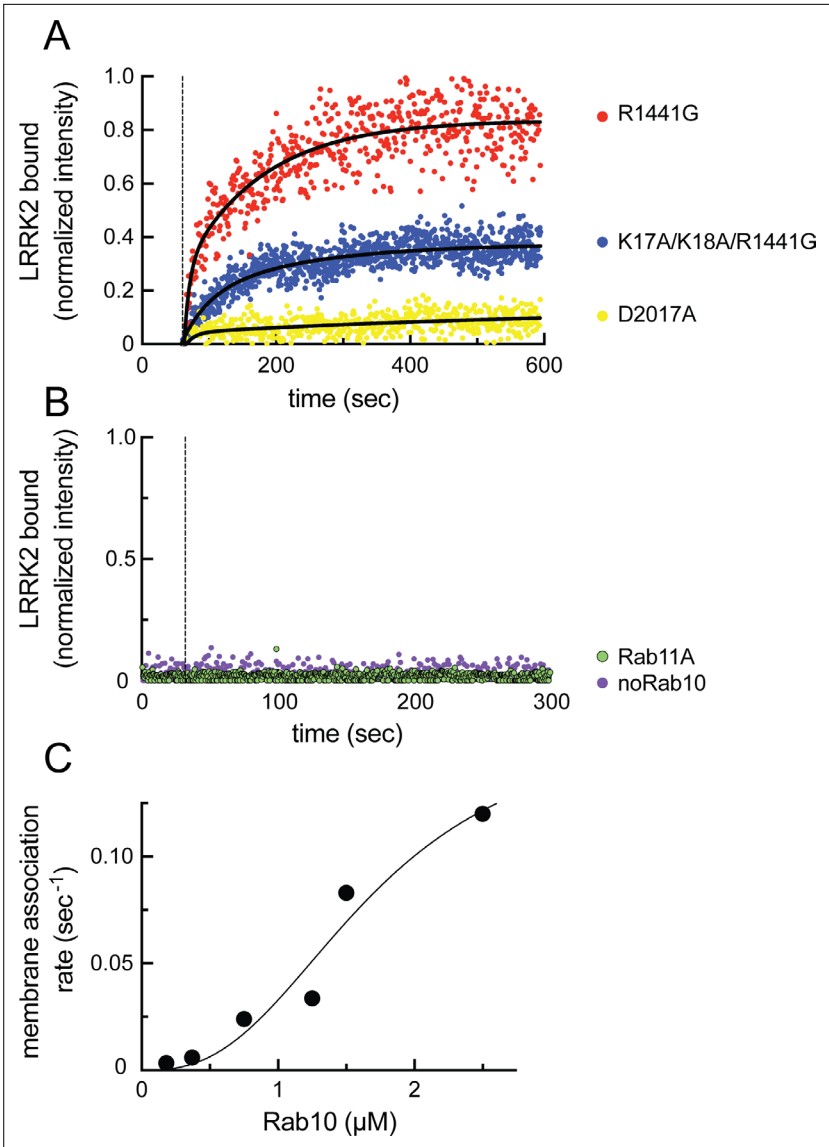

**Figure 8.** Feed-forward pathway for Rab10 phosphorylation is dependent on LRRK2 kinase activity.
(**A**) Fluorescence intensity traces of individual, single molecules of 7 nM CF633-labeled FLAG-LRRK2 R1441G on a substrate-supported lipid bilayer decorated with lipid-anchored GFP-Rab10 Q68L-His across 600 s of live total internal reflection (TIRF) microscopy. Red, R1441G; blue, K17A/K18A/R1441G; yellow, D2017A. (**B**) Reactions were carried out as in (**A**) except Rab10 was omitted (purple) or Rab10 was replaced with Rab11 (green). Dashed lines in (**A**) and (**B**) represent time of addition of fluorescently labeled LRRK2 at 60 s; shown are representative experiments carried out at least three times for each condition. Fluorescence intensity was fitted by a nonlinear regression curve for two-phase association. Fold change was calculated by dividing the average fluorescence intensity at steady state and subtracting background fluorescence intensity average determined from 60 s prior to LRRK2 addition. (**C**) Rate of membrane association of LRRK2 as a function of Rab10 concentration. This curve was fitted by a nonlinear regression fit using PRISM software (MathWorks) to determine a Hill coefficient. Data are from two independent experiments plotted together.

The online version of this article includes the following source data and figure supplement(s) for figure 8:

**Figure supplement 1.** Quantitative analysis of total internal reflection (TIRF) images of LRRK2 recruitment on planar lipid bilayers.

**Figure supplement 2.** The LRRK2 Armadillo domain can bind phosphorylated Rab10 and unphosphorylated Rab8A simultaneously.

**Figure supplement 2—source data 1.** Raw data for gels.

**Figure supplement 2—source data 2.** Annotated gels.

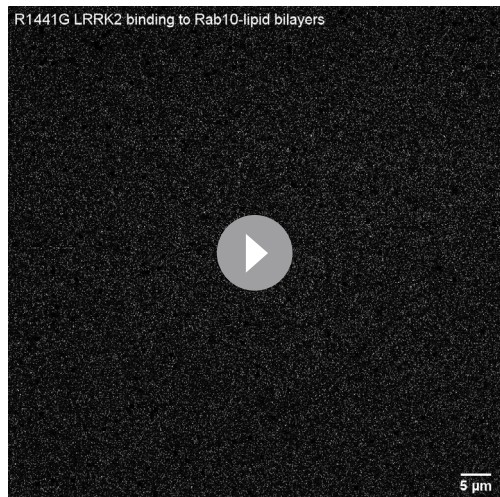

**Video 1.** Total internal reflection (TIRF) microscopy of R1441G LRRK2 binding to Rab10-lipid bilayers. Captured at 1 frame/s and compressed 20×.
https://elifesciences.org/articles/79771/figures#video1

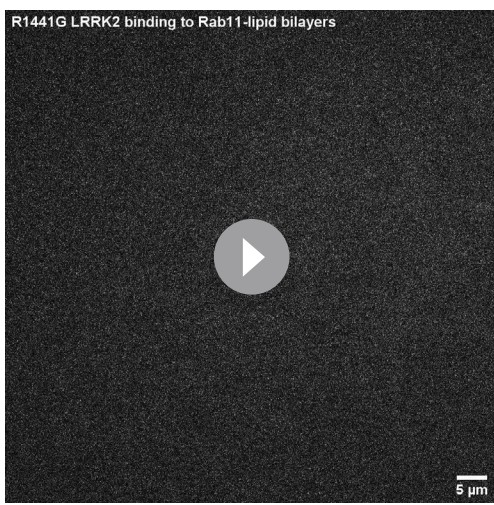

**Video 3.** Total internal reflection (TIRF) microscopy of R1441G LRRK2 binding to Rab11-lipid bilayers. Captured at 0.5 frame/s and compressed 40×.
https://elifesciences.org/articles/79771/figures#video3

LRRK2 is difficult to dye-label mono-molecularly as the N-terminus is engaged in phosphoRab binding and the C-terminus is critical for activity. Nevertheless, analysis of the distribution of single-molecule fluorescence intensity of our CF633-labeled LRRK2 preparation revealed a sharp peak, whether the preparation was evaluated immediately upon binding to Rab10 on bilayers (*Figure 8—figure supplement 1A,B, and D*) or when spotted onto poly-lysine-coated glass (*Figure 8—figure supplement 1B*, far-right column). *Figure 8—figure supplement 1A and B* show the intensity at time t for large numbers of fluorescent molecules, either over 500 s (A) or 30 s (B). The intensity shift over time (*Figure 8—figure supplement 1A and B*) may imply that the molecules slowly dimerize with a half-time of 100–200 s, but additional work would be needed to confirm this. Continuous traces of the 30 longest lived spots showed that for some events this increase occurs even more quickly (*Figure 8—figure supplement 1C*). The fluorescent molecules remain on the bilayers for a significant period of time (*Figure 8—figure supplement 1D,E*); moreover, when the molecules first bind to the surface, the single-peak distribution of intensity

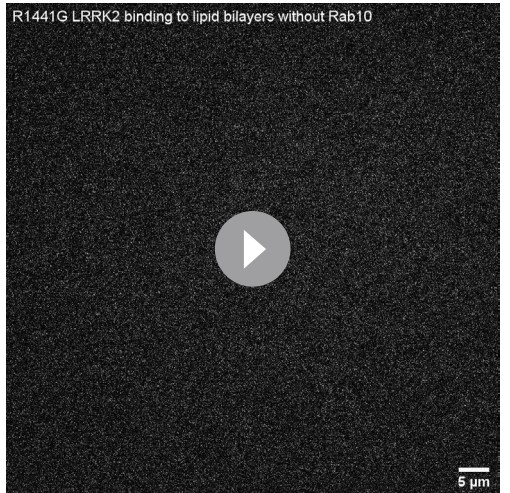

**Video 2.** Total internal reflection (TIRF) microscopy of R1441G LRRK2 binding to lipid bilayers without Rab10. Captured at 1 frame/s and compressed 20×.
https://elifesciences.org/articles/79771/figures#video2

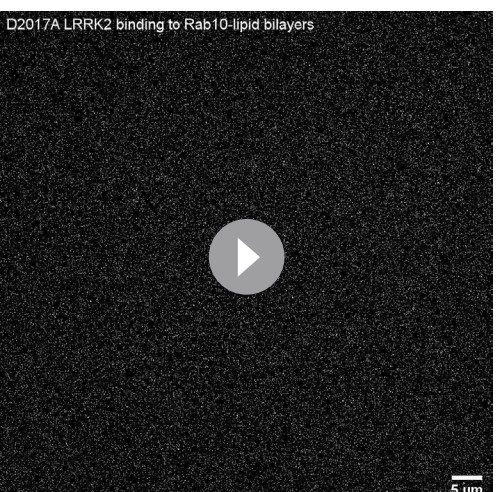

**Video 4.** Total internal reflection (TIRF) microscopy of D2017A LRRK2 binding to Rab10-lipid bilayers. Captured at 1 frame/s and compressed 20×.
https://elifesciences.org/articles/79771/figures#video4

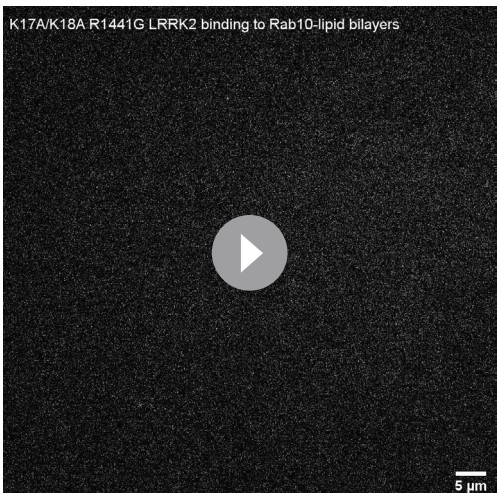

**Video 5.** Total internal reflection (TIRF) microscopy of K17A/K18A/R1441G LRRK2 binding to Rab10-lipid bilayers. Captured at 0.5 frame/s and compressed 40×. https://elifesciences.org/articles/79771/figures#video5

does not change, irrespective of the time during the experiment that it actually binds (*Figure 8—figure supplement 1D*). This gives us confidence that any changes observed were not occurring in solution and require Rab engagement. Note that we detect a minor species at $\log_2 = 2.5$ that constitutes between 2 and 6% of the molecules (*Figure 8—figure supplement 1D and F*); this may represent dual-labeled proteins and/or rare tetrameric complexes.

To confirm that LRRK2 Armadillo domain can bind both non-phosphorylated and phosphorylated Rabs simultaneously, GST-Rab8A was immobilized on glutathione agarose and Armadillo domain (1–552) protein pre-bound. Purified, phosphoRab10 was then added, and immunoblotting showed that phosphoRab10 bound to the beads only in the presence of Rab8A-anchored, Armadillo fragment (*Figure 8—figure supplement 2*). Thus, simultaneous Rab binding at both sites #1 and #2 can occur, and is predicted to increase avidity of LRRK2 membrane association, consistent with our membrane recruitment data.

## PhosphoRab8 activates LRRK2 phosphorylation of Rab10 protein

The data presented thus far are consistent with apparent activation of LRRK2 by cooperative recruitment of the kinase to membrane microdomains enriched in Rab protein substrates. It was formally possible, however, that phosphoRab binding actually activates the kinase itself. To test this, we monitored the generation of phosphoRab10 using a highly specific anti-phosphoRab10 monoclonal antibody in conjunction with immunoblotting. Rab10 protein was then phosphorylated by purified, full-length LRRK2 kinase in vitro, with and without addition of pre-phosphorylated Rab8A protein. As shown in *Figure 9A and C*, the presence of stoichiometrically phosphorylated Rab8A (*Dhekne et al., 2021*) stimulated the rate of in vitro Rab10 phosphorylation by approximately fourfold. Importantly, the ability of phosphoRab8A to stimulate LRRK2-mediated Rab10 phosphorylation required LRRK2's K18 that is needed for phosphoRab binding (*Figure 9B and D*). We speculate that phosphoRab binding to the absolute N-terminus influences LRRK2's higher-order structure to stimulate kinase activity.

## Discussion

LRRK2 is ~90% cytosolic (cf. *Purlyte et al., 2018*), and little was known about why membrane-associated LRRK2 appears to be much more active than the cytosolic pool of kinase. We have confirmed here that LRRK2 kinase relies upon substrate Rab GTPases to achieve membrane association and revealed that LRRK2 utilizes two distinct Rab-binding sites within its N-terminal Armadillo domain for this purpose. Site #1 (*Figure 5B*) binds multiple, non-phosphorylated Rab substrates including Rab8A, Rab10, and Rab29, as well as the highly tissue-specific and non-substrate, Rab29-related, Rab32 and Rab38 proteins (*Waschbüsch et al., 2014*; *McGrath et al., 2021*). The second site (#2) is located at LRRK2's absolute N-terminus at a significant distance from the kinase active site; this site shows strong preference for phosphorylated Rab8A and Rab10 proteins. Our data show that both sites can be occupied simultaneously.

*Figure 10* shows our current model for LRRK2 membrane recruitment. LRRK2 will interact reversibly with any one of the subset of Rab proteins that can bind to site #1. Rab29 shows the highest affinity for this site, but Rab8A can also bind with physiologically relevant affinity and is much more abundant in cells. Rab GTPases cluster in microdomains on distinct membrane surfaces (*Pfeffer, 2017*; *Sönnichsen et al., 2000*; *de Renzis et al., 2002*; *Barbero et al., 2002*), thus this initial LRRK2

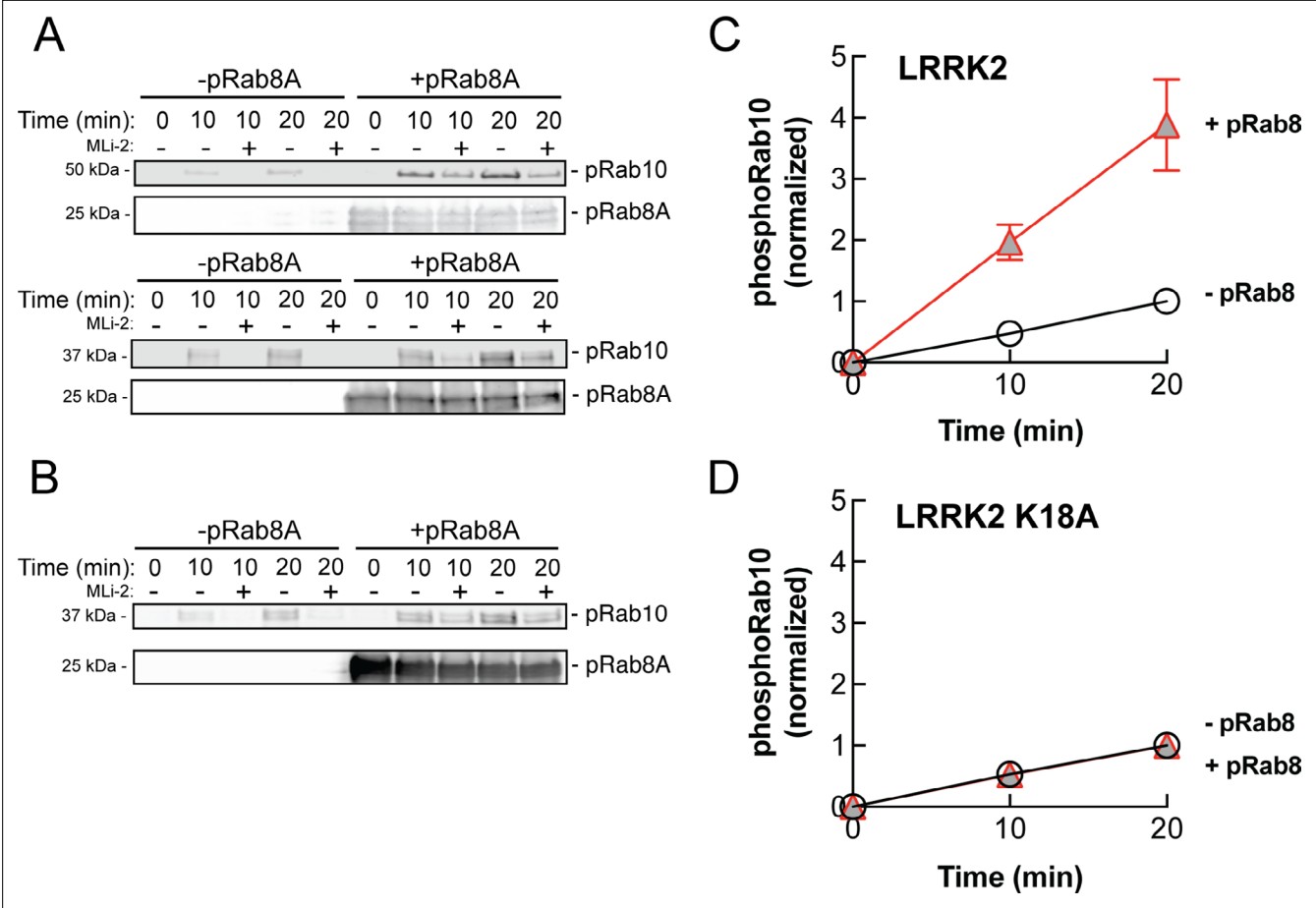

**Figure 9.** PhosphoRab8A activates LRRK2 phosphorylation of Rab10 in solution. (**A**) Immunoblot analysis of the kinetics of LRRK2 G2019S phosphorylation of Rab10 with and without additional pRab8. Upper gel: GFP-Rab10 Q68L His substrate. Lower gel: His-Sumo-Rab10 wild-type full-length substrate. Indicated reactions contained 200 nM MLi-2. pRab8A was detected with anti-phosphoRab8A antibody. (**B**) Same as panel (**A**) with K18A-LRRK2-R1441G and His-Sumo-Rab10 wild-type full-length as substrate. PhosphoRab8A was detected with total Rab8 antibody. (**C**) Kinetics of phosphoRab10 production as in (**A**). Shown are the combined means of independent, quadruplicate determinations ± SEM, as indicated. (**D**) PhosphoRab10 production as in (**B**). Shown are the combined means of independent duplicate determinations,± SEM, as indicated. Background signal in the presence of pRab8A is likely due to trace MST3 contamination that is not sensitive to MLi-2 inhibition and was subtracted. pRab8 preparation was by method #1 for (**A**), upper gel, and (**B**), and method #2 was used in panel (**A**), lower gel.

The online version of this article includes the following source data for figure 9:

**Source data 1.** Raw data for gels.

**Source data 2.** Annotated gels.

membrane association will bring the kinase in contact with other copies of the same substrate Rab proteins for phosphorylation. After an initial phosphorylation event, LRRK2 will then be held in place by bivalent association with one phosphorylated and one non-phosphorylated Rab protein. By binding to the kinase reaction product, LRRK2 enhances its effective, local activity by increasing the probability with which it will encounter another substrate Rab protein.

Despite relatively similar affinities for their respective Rab-binding partners, the phosphoRab-specific site appears to drive stable LRRK2 membrane association as mutation of two key lysine residues strongly impacts co-localization of LRRK2 protein with phosphoRabs in cells. In addition, kinase activity leads to a much higher degree of LRRK2 association with planar lipid bilayers despite the presence of binding site #1 for non-phosphorylated Rabs. Finally, K17/K18A LRRK2 that cannot bind to phosphorylated Rab proteins showed lower bilayer association in comparison with native LRRK2, confirming the importance of this interaction. LRRK2 phosphorylation of Rab GTPases is therefore required to form a new, additional interaction interface that greatly enhances the overall avidity of LRRK2 membrane association.

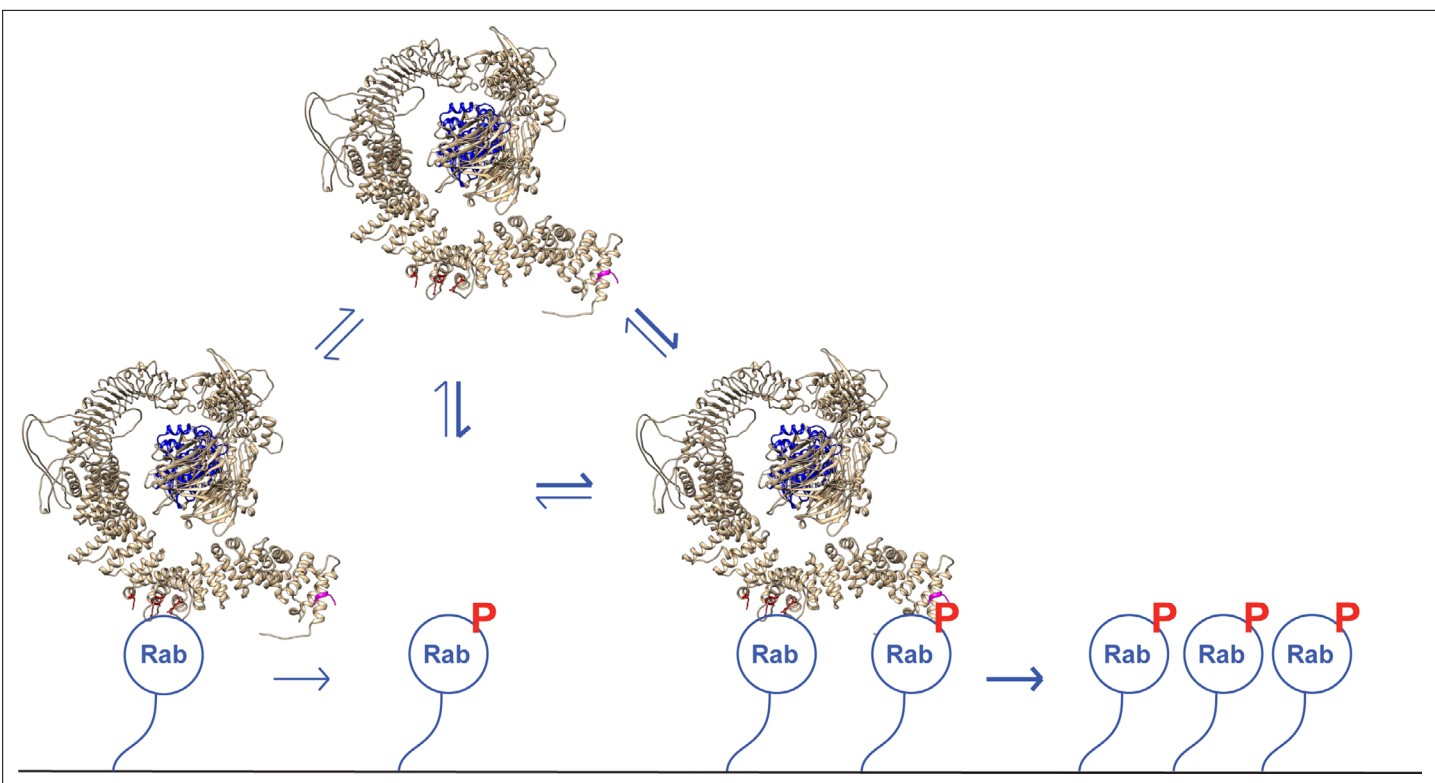

**Figure 10.** A model for LRRK2 membrane recruitment. LRRK2 can interact with non-phosphorylated Rab GTPases via site #1. Once membrane bound, it can generate phosphoRabs that can now engage site #2. Rab binding to both sites increases the avidity of LRRK2 for membranes and retains LRRK2 on the membrane surface to phosphorylate more Rab substrates. We have shown that LRRK2 binding to phosphoRabs also activates the kinase, likely by altering its oligomeric state.

We also discovered that phosphoRab8A stimulates LRRK2 kinase action on Rab10 protein. We were not able to test the reverse scenario as the phosphoRab8A antibody is not adequately specific and cross-reacts with phosphoRab10 protein. Nevertheless, it seems very likely that phosphoRab10 will also activate LRRK2 for other substrate phosphorylation events. The most likely explanation is that phosphoRab binding to the LRRK2 N-terminus encourages an overall enzyme architecture that favors the active conformation. LRRK2 assumes multiple oligomeric states, and phosphoRab engagement and/or dual Rab engagement of the Armadillo domain likely influences the overall architecture of the enzyme.

It is important to note that quantitative mass spectrometry indicates that Rab10 is present at ~600 times the copy number as LRRK2 in MEF cells and brain tissue (https://copica.proteo.info/#/copy-browse). Thus, if Rab10 is assumed to exist in cells at ~2–5 µM (*Itzhak et al., 2016*), LRRK2 will be present overall at about 3–8 nM. These are very close to the concentrations used in our in vitro reconstitution experiments. Future experiments will be needed to elucidate the precise molecular state of LRRK2 upon engagement with Rab GTPases at sites #1 and #2.

*Nichols et al., 2007* reported a single family with two affected siblings harboring LRRK2 E10K mutations. These patients presented with classic Parkinson's disease symptoms at age 57 including bradykinesia, muscular rigidity, postural instability, and resting tremor. Compared with 46 G2019S LRRK2 patients in that study whose disease onset was on average, 63.5 years, the two siblings had a more severely disabling disease, as indicated by a higher Hoehn and Yahr assessment score (4 vs. 2.5, where 5 represents confinement to bed or wheelchair unless aided). Our study provides a molecular explanation for how a mutation located far from the kinase or ROC-COR domains may cause Parkinson's disease. We predict that the E10K mutation increases LRRK2 phosphoRab binding and membrane association and may display an even higher apparent activity than the most common pathogenic G2019S mutation. This distinction would need to be evaluated under conditions of MLi-2 washout as exogenous expression would mask this subtle mechanistic feature.

The ability of multiple Rab-binding sites to anchor LRRK2 on membranes will make the kinase appear more active than the pool of cytosolic LRRK2 protein. Rab binding may also increase access of LRRK2 to other kinases that stabilize it in a more active conformation. Anchoring LRRK2's N-terminus may also influence autophosphorylation, which could also drive LRRK2 towards a more catalytically active conformation. Future structural studies of membrane-anchored LRRK2 will provide important, additional information related to all of these possibilities.

# Materials and methods

**Key resources table**

| Reagent type (species) or resource | Designation | Source or reference | Identifiers | Additional information |
|---|---|---|---|---|
| Antibody | Anti-LRRK2 (mouse monoclonal) | NeuroMab RRID:AB_2877351 | N241A/34 | (1:1000) |
| Antibody | Anti-LRRK2 phospho S935 (rabbit monoclonal) | Abcam RRID:AB_2904231 | UDD2 | (1:1000) |
| Antibody | Anti-Rab10 (mouse monoclonal) | Nanotools RRID:AB_2921226 | 0680-100/Rab10-605B11 | (1:1000) |
| Antibody | Anti-Rab10 (phospho T73) (rabbit monoclonal) | Abcam RRID:AB_2811274 | ab230261 | (1:1000) |
| Antibody | Anti-FLAG M2 (mouse monoclonal) | MilliporeSigma RRID:AB_262044 | F-1804 | (1:2000) |
| Strain, strain background (*Escherichia coli*) | *E. coli* DH5α | Thermo Fisher | 18258012 | |
| Strain, strain background (*E. coli*) | *E. coli* STBL3 | Thermo Fisher | C737303 | |
| Strain, strain background (*E. coli*) | *E. coli* Rosetta DE3 pLys | Millipore | 70956 | |
| Cell line (*Homo sapiens*) | HeLa | ATCC | CCL-2 | |
| Cell line (*H. sapiens*) | HEK293T | ATCC | CRL-3216 | |
| Chemical compound, drug | MLi-2 | MRC PPU | | |
| Chemical compound, drug | Creatine phosphate | Fluka Analytical | #27920 | 20 mM |
| Commercial assay or kit | RED-NHS 2nd Generation (Amine Reactive) Protein Labeling Kit | NanoTemper Technologies | MO-L011 | |
| Commercial assay or kit | CF 633 Succinimidyl Ester Protein Labeling Kit | Biotium | #92217 | |
| Other | Creatine Phosphokinase | Sigma | C3755 | 30U |
| Chemical compound, drug | 18:1 (Δ9-Cis) PC (DOPC) | Avanti Polar Lipids | #850375 | 11 μmol |
| Chemical compound, drug | 18:1 PS (DOPS) | Avanti Polar Lipids | #840035 | 5 μmol |
| Chemical compound, drug | 18:1 DGS-NTA(Ni) | Avanti Polar Lipids | #790404 | 0.85 μmol |
| Chemical compound, drug | 18:1 PI(4)P | Avanti Polar Lipids | #850151 | 0.15 μmol |
| Chemical compound, drug | DiD | Thermo Fisher | D7757 | 0.01 μmol |
| Recombinant DNA reagent | pNIC Bsa-4 His-Sumo Rab10 Q68L 1–181 | Gift of Amir Khan | | Human |
| Recombinant DNA reagent | pET15b His-Mst3 | Gift of Amir Khan | | Human |
| Recombinant DNA reagent | pET21b GFP-Rab10 Q68L-His | Addgene RRID:Addgene_186015 | 186015 | Human |
| Recombinant DNA reagent | pET21b His Rab8A Q67L | Addgene RRID:Addgene_186014 | 186014 | Human |

*Continued on next page*

*Continued*

| Reagent type (species) or resource | Designation | Source or reference | Identifiers | Additional information |
|---|---|---|---|---|
| Recombinant DNA reagent | pQE-80L 2xHis-Rab29 | Addgene RRID:Addgene_186021 | 186021 | Human |
| Recombinant DNA reagent | pGEB GST-Rab8A-Q67L | Addgene RRID:Addgene_86079 | 86079 | Human |
| Recombinant DNA reagent | His-Rab11 | Gift of Marino Zerial | | Canine |
| Recombinant DNA reagent | pQE-80L 2xHis-LRRK2 Armadillo 1–552 | Addgene RRID:Addgene_186017 | 186017 | Human |
| Recombinant DNA reagent | pQE-80L 2xHis-LRRK2-Armadillo 1–159 | Addgene RRID:Addgene_186016 | 186016 | Human |
| Recombinant DNA reagent | pQE-80L 2xHis-LRRK2-Armadillo 350–550 | Addgene RRID:Addgene_186018 | 186018 | Human |
| Recombinant DNA reagent | pQE-80L 2xHis-LRRK2-Armadillo K17A | Addgene RRID:Addgene_186019 | 186019 | Human |
| Recombinant DNA reagent | pQE-80L 2xHis-LRRK2-Armadillo K18A | Addgene RRID:Addgene_186020 | 186020 | Human |
| Recombinant DNA reagent | pCMV5 FLAG-LRRK2 K17A/K18A/R1441G | Addgene RRID:Addgene_186012 | 186012 | Human |
| Recombinant DNA reagent | pCMV5 FLAG-LRRK2 | MRC PPU Reagents and Services, University of Dundee ('MRC PPU') | DU6841 | Human |
| Recombinant DNA reagent | pCMV5 FLAG-LRRK2 R1441G | MRC PPU | DU13077 | Human |
| Recombinant DNA reagent | pCMV5 FLAG-LRRK2 D2017A | MRC PPU | DU52725 | Human |
| Recombinant DNA reagent | pcDNA5D FRT TO GFP LRRK2 WT | MRC PPU | DU13363 | Human |
| Recombinant DNA reagent | pcDNA5D FRT TO GFP LRRK2 R361E | MRC PPU | DU62605 | Human |
| Recombinant DNA reagent | pcDNA5D FRT TO GFP LRRK2 D392K | MRC PPU | DU72261 | Human |
| Recombinant DNA reagent | pcDNA5D FRT TO GFP LRRK2 R399E | MRC PPU | DU72262 | Human |
| Recombinant DNA reagent | pcDNA5D FRT TO GFP LRRK2 L403A | MRC PPU | DU72263 | Human |
| Recombinant DNA reagent | pcDNA5D FRT TO GFP LRRK2 L406A | MRC PPU | DU72266 | Human |
| Recombinant DNA reagent | pcDNA5D FRT TO GFP LRRK2 M407A | MRC PPU | DU72267 | Human |
| Recombinant DNA reagent | pcDNA5D FRT TO GFP LRRK2 K439E | MRC PPU | DU72268 | Human |
| Recombinant DNA reagent | pcDNA5D FRT TO GFP LRRK2 L443A | MRC PPU | DU72270 | Human |
| Recombinant DNA reagent | pcDNA5D FRT TO GFP LRRK2 K451E | MRC PPU | DU72271 | Human |
| Recombinant DNA reagent | pcDNA5D FRT TO GFP LRRK2 D478Y | MRC PPU | DU68605 | Human |
| Recombinant DNA reagent | pcDNA5D FRT TO GFP LRRK2 D2017A | MRC PPU | DU13364 | Human |
| Recombinant DNA reagent | pcDNA5D FRT TO GFP LRRK2 R1441C | MRC PPU | DU13387 | Human |
| Recombinant DNA reagent | pcDNA5D FRT TO GFP LRRK2 R1441C R361E | MRC PPU | DU72304 | Human |
| Recombinant DNA reagent | pcDNA5D FRT TO GFP LRRK2 R1441C D392K | MRC PPU | DU72305 | Human |
| Recombinant DNA reagent | pcDNA5D FRT TO GFP LRRK2 R1441C R399E | MRC PPU | DU72306 | Human |
| Recombinant DNA reagent | pcDNA5D FRT TO GFP LRRK2 R1441C L403A | MRC PPU | DU72307 | Human |

*Continued*

| Reagent type (species) or resource | Designation | Source or reference | Identifiers | Additional information |
|---|---|---|---|---|
| Recombinant DNA reagent | pcDNA5D FRT TO GFP LRRK2 R1441C L406A | MRC PPU | DU72308 | Human |
| Recombinant DNA reagent | pcDNA5D FRT TO GFP LRRK2 R1441C M407A | MRC PPU | DU72309 | Human |
| Recombinant DNA reagent | pcDNA5D FRT TO GFP LRRK2 R1441C K439E | MRC PPU | DU72310 | Human |
| Recombinant DNA reagent | pcDNA5D FRT TO GFP LRRK2 R1441C L443A | MRC PPU | DU72311 | Human |
| Recombinant DNA reagent | pcDNA5D FRT TO GFP LRRK2 R1441C K451E | MRC PPU | DU72312 | Human |

| Reagent type (species) or resource | Designation | Source or reference | Identifiers | Additional information |
|---|---|---|---|---|
| Recombinant DNA reagent | pCMV5D HA RAB29 | MRC PPU | DU50222 | Human |
| Recombinant DNA reagent | pcDNA5D FRT TO GFP LRRK2 1–950 | MRC PPU | DU62702 | Human |
| Recombinant DNA reagent | pcDNA5D FRT TO GFP LRRK2 1–900 | MRC PPU | DU62701 | Human |
| Recombinant DNA reagent | pcDNA5D FRT TO GFP LRRK2 1–850 | MRC PPU | DU62700 | Human |
| Recombinant DNA reagent | pcDNA5D FRT TO GFP LRRK2 1–800 | MRC PPU | DU62693 | Human |
| Recombinant DNA reagent | pcDNA5D FRT TO GFP LRRK2 1–750 | MRC PPU | DU62726 | Human |
| Recombinant DNA reagent | pcDNA5D FRT TO GFP LRRK2 1–700 | MRC PPU | DU62689 | Human |
| Recombinant DNA reagent | pcDNA5D FRT TO GFP LRRK2 1–650 | MRC PPU | DU62678 | Human |
| Recombinant DNA reagent | pcDNA5D FRT TO GFP LRRK2 1–600 | MRC PPU | DU62677 | Human |
| Recombinant DNA reagent | pcDNA5D FRT TO GFP LRRK2 1–550 | MRC PPU | DU62676 | Human |
| Recombinant DNA reagent | pcDNA5D FRT TO GFP LRRK2 1–500 | MRC PPU | DU62675 | Human |
| Recombinant DNA reagent | pcDNA5D FRT TO GFP LRRK2 50–1000 | MRC PPU | DU62725 | Human |
| Recombinant DNA reagent | pcDNA5D FRT TO GFP LRRK2 100–1000 | MRC PPU | DU62742 | Human |
| Recombinant DNA reagent | pcDNA5D FRT TO GFP LRRK2 150–1000 | MRC PPU | DU62674 | Human |
| Recombinant DNA reagent | pcDNA5D FRT TO GFP LRRK2 200–1000 | MRC PPU | DU62679 | Human |
| Recombinant DNA reagent | pcDNA5D FRT TO GFP LRRK2 250–1000 | MRC PPU | DU62680 | Human |
| Recombinant DNA reagent | pcDNA5D FRT TO GFP LRRK2 300–1000 | MRC PPU | DU62681 | Human |
| Recombinant DNA reagent | pcDNA5D FRT TO GFP LRRK2 350–1000 | MRC PPU | DU62682 | Human |
| Recombinant DNA reagent | pcDNA5D FRT TO GFP LRRK2 400–1000 | MRC PPU | DU62683 | Human |
| Recombinant DNA reagent | pcDNA5D FRT TO GFP LRRK2 450–1000 | MRC PPU | DU62684 | Human |
| Recombinant DNA reagent | pcDNA5D FRT TO GFP LRRK2 500–1000 | MRC PPU | DU62685 | Human |
| Recombinant DNA reagent | pcDNA5D FRT TO GFP LRRK2 550–1000 | MRC PPU | DU62686 | Human |
| Recombinant DNA reagent | pcDNA5D FRT TO GFP LRRK2 600–1000 | MRC PPU | DU62687 | Human |
| Recombinant DNA reagent | pcDNA5D FRT TO GFP LRRK2 350–550 | MRC PPU | DU68397 | Human |
| Recombinant DNA reagent | pcDNA5D FRT TO GFP LRRK2 350–500 | MRC PPU | DU68398 | Human |

Continued

| Reagent type (species) or resource | Designation | Source or reference | Identifiers | Additional information |
|---|---|---|---|---|
| Recombinant DNA reagent | His-SUMO Rab10 | MRC PPU | DU51062 | Human |
| Recombinant DNA reagent | His Rab7 | Gift of Marino Zerial | | |
| Software, algorithm | Fiji | PMID:29187165 | RRID:SCR_002285 | |
| Software, algorithm | CellProfiler | PMID:29969450 | RRID:SCR_007358 | |
| Software, algorithm | TrackIt | PMID:33947895 | | |
| Software, algorithm | Chimera 2 | PMID:15264254 | RRID:SCR_004097 | |
| Software, algorithm | ChimeraX | PMID:32881101 | RRID:SCR_015872 | |
| Software, algorithm | NanoTemper NTAAffinityAnalysis | MO.Affinity Analysis v2.2.5 | | |
| Software, algorithm | Prism | Prism 9 version 9.3.1 (350) | RRID:SCR_002798 | |
| Software, algorithm | R CRAN R package | Version 4.2.0 (2022-04-22) | RRID:SCR_003005 | |
| Software, algorithm | Dplyr_1.0.9 | | RRID:SCR_016708 | |
| Software, algorithm | ggridges_0.5.3 | | | |
| Software, algorithm | ggplot_3.3.6 | | RRID:SCR_014601 | |

## Cloning and plasmids

DNA constructs were amplified in *Escherichia coli* DH5α or STBL3 and purified using mini prep columns (EconoSpin). DNA sequence verification was performed by Sequetech (http://www.sequetech.com). pNIC Bsa-4 His-Sumo Rab10 Q68L 1–181 and pET15b His-Mst3 were kind gifts of Amir Khan (Harvard University). pET21b GFP-Rab10 Q68L-His was subcloned from GFP-Rab10 (*Gomez et al., 2019*) into pET21b. The C-terminal His-tagged version was generated by Gibson assembly. His Rab8A Q67L was subcloned from HA-Rab8A (DU35414, Medical Research Council at Dundee) into pET14b. Point mutations were generated using site-directed mutagenesis. His-Rab29 wild type was subcloned from HA-Rab29 (DU5022, Medical Research Council at Dundee) into the pQE-80L backbone. pCMV5 FLAG-LRRK2 (DU6841), Flag-LRRK2 R1441G (DU13077), His-SUMO Rab10 (DU51062), and FLAG-LRRK2 D2017A (DU52725) were obtained from the Medical Research Council at Dundee. His-Armadillo 1–552, 1–159, and 350–550 were all cloned from pCMV5 FLAG-LRRK2 into pQE-80L. K17A, K18A, and K17A/K18A LRRK2 and LRRK2 Armadillo were generated using site-directed mutagenesis. All cloning and subcloning were done by Gibson assembly.

## Rab GTPase, LRRK2 Armadillo domain, and LRRK2 purification

His Rab29, His Rab10 Q68L (1–181), His Rab10 Q68L (full length) His, His-Mst3, His-Rab8A Q67L, His-LRRK2 Armadillo (1–552), His-LRRK2 Armadillo (1–159), His-LRRK2 Armadillo (350–550), His-LRRK2 Armadillo K17A, His-LRRK2 Armadillo K18A, and GST-Rab8A Q67L were purified after expression in *E. coli* BL21 (DE3 pLys). Detailed protocols can be found in *Gomez et al., 2020* (https://dx.doi.org/10.17504/protocols.io.bffrjjm6) and *Vides and Pfeffer, 2021* (https://dx.doi.org/10.17504/protocols.io.bvvmn646). Bacterial cells were grown at 37°C in Luria Broth and induced at A600 nm = 0.6–0.7 by the addition of 0.3 mM isopropyl-1-thio-β-ᴅ-galactopyranoside (Gold Biotechnology) and harvested after 18 hr at 18°C. The cell pellets were resuspended in ice-cold lysis buffer (50 mM HEPES, pH 8.0, 10% [vol/vol] glycerol, 500 mM NaCl, 10 mM imidazole [for His-tagged purification only], 5 mM $MgCl_2$, 0.2 mM tris(2-carboxyethyl) phosphine [TCEP], 20 μM GTP, and EDTA-free protease inhibitor cocktail [Roche]). The resuspended bacteria were lysed by one passage through an Emulsiflex-C5 apparatus (Avestin) at 10,000 lbs/in² and centrifuged at 40,000 rpm for 45 min at 4°C in a Beckman Ti45 rotor. Cleared lysate was filtered through a 0.2 μm filter (Nalgene) and passed over a HiTrap TALON crude 1 mL column (Cytiva) for His-tagged proteins or a GSTrap High Performance 1 mL column (Cytiva) for GST-tagged proteins. The column was washed with lysis buffer until absorbance values reached pre-lysate values. Protein was eluted with a gradient from 20 to 500 mM imidazole

containing lysis buffer for His-tagged proteins or 0–50 mM reduced glutathione containing lysis buffer for GST-tagged proteins. Peak fractions analyzed by 10% SDS-PAGE to locate protein. The eluate was buffer exchanged and further purified by gel filtration on Superdex-75 (GE Healthcare) with 50 mM HEPES, pH 8, 5% (vol/vol) glycerol, 150 mM NaCl, 5 mM MgCl$_2$, 0.1 mM tris(2-carboxyethyl) phosphine (TCEP), and 20 µM GTP.

LRRK2 R1441G was transfected into HEK293T cells with Polyethylenimine HCl MAX 4000 (PEI) (Polysciences, Inc) and purified 48 hr post transfection. Cells were lysed in 50 mM HEPES pH 8, 150 mM NaCl, 1 mM EDTA, 0.5% Triton-X 100, 10% (vol/vol) glycerol and protease inhibitor cocktail (Roche). Lysate was centrifuged at 15,000 × *g* for 20 min in Fiberlite F15 rotor (Thermo Fisher). Clarified lysate was filtered through 0.2 µm syringe filters and circulated over anti-FLAG M2 affinity gel (Sigma) at 4°C for 4 hr using a peristaltic pump. The affinity gel was washed with 6-column volumes of lysis buffer followed by 6-column volumes of elution buffer (50 mM HEPES pH 8, 150 mM NaCl, and 10% [vol/vol] glycerol). Protein was eluted from resin with 5-column volumes of FLAG peptide (0.25 mg/mL) containing elution buffer. Eluate was supplemented to 20 µM GTP, 1 mM ATP, and 2 mM MgCl$_2$.

## In vitro Rab phosphorylation and microscale thermophoresis

A detailed method can be found at https://dx.doi.org/10.17504/protocols.io.bvvmn646. His-Rab10 Q68L 1–181 or His-Rab8A Q67L was incubated with His-Mst3 kinase at a molar ratio of 3:1 (substrate:kinase). The reaction buffer was 50 mM HEPES, pH 8, 5% (vol/vol) glycerol, 100 mM NaCl, 5 mM MgCl$_2$, 0.2 mM TCEP, 20 µM GTP, 5 µM BSA, 0.01% Tween-20, and 2 mM ATP (no ATP for negative control). The reaction mixture was incubated at 27°C for 30 min in a water bath. Phosphorylation completion was assessed by Western blot of Phos-tag gels. Immediately after phosphorylation, the samples were transferred to ice before LRRK2 Armadillo domain binding determination. See also (*Knebel et al., 2021*); https://dx.doi.org/10.17504/protocols.io.bvjxn4pn.

Protein–protein interactions were monitored by microscale thermophoresis using a Monolith NT.115 instrument (NanoTemper Technologies). His LRRK2 Armadillo (1–552), (1–159), (350–550), K17A and K18A were labeled using RED-NHS 2nd Generation (Amine Reactive) Protein Labeling Kit (NanoTemper Technologies). For all experiments, the unlabeled protein partner was titrated against a fixed concentration of the fluorescently labeled LRRK2 Armadillo (100 nM); 16 serially diluted titrations of the unlabeled protein partner were prepared to generate one complete binding isotherm. Binding was carried out in a reaction buffer in 0.5 mL Protein LoBind tubes (Eppendorf) and allowed to incubate in the dark for 30 min before loading into NT.115 premium treated capillaries (NanoTemper Technologies). A red LED at 30% excitation power (red filter, excitation 605–645 nm, emission 680–685 nm) and IR-laser power at 60% was used for 30 s followed by 1 s of cooling. Data analysis was performed with NTAffinityAnalysis software (NanoTemper Technologies) in which the binding isotherms were derived from the raw fluorescence data and then fitted with both NanoTemper software and GraphPad Prism to determine the $K_D$ using a nonlinear regression method. The binding affinities determined by the two methods were similar. Shown are averaged curves of Rab GTPase-binding partners from single readings from two different protein preparations. Note that the affinities reported here are underestimates as preps of His Rab10-Q68L (1–181) and His-Rab8A Q67L routinely contained a 50:50 ratio of bound GTP:GDP as determined by mass spectroscopy; data were not corrected for this.

## Cell culture and immunoblotting

HEK293T and HeLa cells were obtained from American Type Culture Collection and were cultured at 37°C and under 5% CO$_2$ in Dulbecco's modified Eagle's medium containing 10% fetal bovine serum, 2 mM glutamine, and penicillin (100 U/mL)/streptomycin (100 µg/mL). HEK293T and HeLa cells were transfected with polyethylenimine HCl MAX 4000 (Polysciences). Cells were routinely checked for Mycoplasma by PCR analysis.

HeLa cells for pRab10 recovery kinetics were lysed 48 hr post transfection and MLi-2 treatment in ice-cold lysis buffer (50 mM Tris pH 7.4, 150 mM NaCl, 0.5% Triton X-100, 5 mM MgCl$_2$, 1 mM sodium orthovanadate, 50 mM NaF, 10 mM 2-glycerophosphate, 5 mM sodium pyrophosphate, 0.1 µg/mL mycrocystin-LR [Enzo Life Sciences], and EDTA-free protease inhibitor cocktail [Sigma-Aldrich]).

Lysates were centrifuged at 14,000 × *g* for 15 min at 4°C and supernatant protein concentrations were determined by Bradford assay (Bio-Rad).

A detailed protocol for blotting is available on protocols.io (*Tonelli and Alessi, 2021*, https://dx. doi.org/10.17504/protocols.io.bsgrnbv6). 20 µg of protein was resolved by SDS-PAGE and transferred onto nitrocellulose membranes using a Bio-Rad Trans-turbo blot system. Membranes were blocked with 2% BSA in Tris-buffered saline with Tween-20 for 30 at room temperature (RT). Primary antibodies used were diluted in blocking buffer as follows: mouse anti-LRRK2 N241A/34 (1:1000, NeuroMab); rabbit anti-LRRK2 phospho S935 (1:1000, Abcam); mouse anti-Rab10 (1:1000, Nanotools); and rabbit anti-phospho Rab10 (1:1000, Abcam). Primary antibody incubations were done overnight at 4°C. LI-COR secondary antibodies diluted in blocking buffer were 680 nm donkey anti-rabbit (1:5000) and 800 nm donkey anti-mouse (1:5000). Secondary antibody incubations were for 1 hr at RT. Blots were imaged using an Odyssey Infrared scanner (LI-COR) and quantified using ImageJ software (*Schneider et al., 2012*).

## MLi-2 washout/pRab10 recovery kinetics

As described by *Ito et al., 2016*, HeLa cell seeded in 6 × 60 mm dishes expressing FLAG-LRRK2, LRRK2 K17A/K18A, LRRK2 R1441G, or LRRK2 R1441G/K17A/K18A for 48 hr were incubated with 200 nm MLi-2 or DMSO for 1 hr under normal growth conditions at 37°C. To remove the MLi-2 inhibitor, cells were washed four times with complete media. Washouts were done to allow for 120–15 min of enzyme activity recovery, after which, cells were harvested.

## Confocal light microscopy

The standard method to obtain images in *Figure 3* and *Figure 3—figure supplements 1–4* can be found on protocols.io (*Purlyte et al., 2022*; https://dx.doi.org/10.17504/protocols.io.b5jhq4j6). For *Figure 6*, cells were plated onto collagen-coated coverslips with indicated plasmids. Cells were washed with ice-cold phosphate-buffered saline (PBS) 3×. Afterward, they were incubated in glutamate buffer (25 mM KCl, 25 mM HEPES pH 7.4, 2.5 mM magnesium acetate, 5 mM EGTA, and 150 mM K glutamate) for 5 min on ice. Coverslips were dipped into liquid nitrogen and held for 5 s before removal. They were thawed at RT, incubated in glutamate buffer for 2 min, and then in PBS for 5 min. Cells were fixed with 3.5% paraformaldehyde in PBS for 15 min, permeabilized for 3 min in 0.1% Triton X-100, and blocked with 1% BSA in PBS. Antibodies were diluted as follows: mouse anti-FLAG (1:2000, Sigma-Aldrich) and rabbit anti pRab10 (1:2000; Abcam). Highly cross-absorbed H+L secondary antibodies (Life Technologies) conjugated to Alexa Fluor 568 or 647 were used at 1:2000. Images were obtained using a spinning disk confocal microscope (Yokogawa) with an electron-multiplying charge-coupled device camera (Andor) and a 100× 1.4 NA oil immersion objective. Mander's correlation coefficients were calculated by analyzing maximum intensity projection images with CellProfiler software (*Stirling et al., 2021*).

Co-localization of Rab29 with full-length LRRK2 and its mutants was quantified using an unbiased CellProfiler pipeline as follows: (1) imported raw .lsm files; (2) metadata extracted from the file headers; (3) images grouped by mutations and split into three channels; (4) nuclei identified as primary objects after rescaling intensities; (5) nucleus is defined as the primary object and cells are identified by 'propagation' as secondary objects; cells are identified as the using the rescaled and smoothened LRRK2 channel; (6). co-localization within whole cells is measured by thresholded (10) Mander's coefficient on the entire batch of images. Data plotted from CellProfiler are relative values.

## Substrate-supported lipid bilayer preparation

A detailed method can be found at dx.doi.org/10.17504/protocols.io.x54v9y7qzg3e/v1. Briefly, we used Lab-TeKII 8 chambered No. 1.5 borosilicate cover glasses (Fisher) for LRRK2 recruitment assays. Reaction chambers were cleaned by 30 min incubation in Piranha solution (1:3 [vol/vol] ratio of 30% $H_2O_2$ and 98% $H_2SO_4$) and extensive washing in Milli-Q water. The reaction chambers were stored in Milli-Q water for up to 2 weeks. Before use, reaction chambers were dried and further cleaned in a Harrick Plasma PDC-32C plasma cleaner for 10 min at 18 W under ambient air.

We prepared substrate-supported lipid bilayers on glass coverslips with 65% DOPC, 29% DOPS, 5% DOGS-NTA[$Ni^2$], 1% PI(4)P, 0.01% DIL (Avanti Polar Lipids; Thermo). The lipid mixture was suspended in 1 mL chloroform and then dried under nitrogen flow in a glass vial and kept under vacuum for

at least 1 hr. The dried lipids were hydrated in SLB buffer (20 mM HEPES pH 8, 150 mM potassium acetate, 1 mM MgCl$_2$) by vortexing to produce multilamellar vesicles (MLVs). SUVs were prepared by bath sonication followed by extrusion through 100 nm polycarbonate membrane 21 times (Avestin). The produced SUVs were stored at –20°C. The supported lipid bilayer was formed in cleaned reaction chambers on glass surfaces by addition of liposomes to a final concentration of 5 mM liposomes in SLB buffer. SUV fusion was induced by addition of 1 mM CaCl$_2$ and incubated for 45 min at 37°C. Next, the unfused vesicles were washed with Milli-Q water and STD buffer (20 mM HEPES pH 8, 150 mM NaCl, 5 mM MgCl$_2$).

Lab-TeKII 8 chambered No. 1.5 borosilicate coverglass (Fisher) were coated with poly-D-lysine as follows (*Adhikari et al., 2022*). 10 mg poly-D-lysine (MPBio # SKU:02150175-CF) was dissolved in 1 mL of sterile Milli-Q water as a 1% stock solution. The stock solution was then diluted twofold in PBS as 1× coating solution. Coating solution (200 µL) was added to the reaction chamber and incubated for 5 min at 37°C. The coating solution was then removed by rinsing the chamber thoroughly with sterile Milli-Q water and equilibrated with reaction buffer (20 mM HEPES pH 8, 150 mM NaCl, 5 mM MgCl$_2$, 4 mM ATP, 20 µM GTP, 20 mM creatine phosphate, 30U creatine phosphokinase) (dx.doi.org/10.17504/protocols.io.x54v9y7qzg3e/v1).

## TIRF microscopy

A detailed method can be found on protocols.io (*Adhikari et al., 2022*; dx.doi.org/10.17504/protocols.io.x54v9y7qzg3e/v1). All LRRK2 recruitment movies were obtained at 25°C at a frame rate capture interval of 1 s using a Nikon Ti-E inverted microscope with the Andor iXon+EMCCD camera model DU885 with PerfectFocus and a Nikon TIRF Apo 100× 1.46 NA oil immersion objective. The imaging was done with 300 EM camera gain and 50 ms exposure time with 200 µW laser intensity. We analyzed the microscopy data with TrackIt (*Kuhn et al., 2021*) to obtain spot density of bound LRRK2.

## Rab10-dependent LRRK2 recruitment

A detailed method can be found on protocols.io (*Adhikari et al., 2022*; dx.doi.org/10.17504/protocols.io.x54v9y7qzg3e/v1). Purified FLAG LRRK2 was labeled with CF633 succinimidyl ester (Biotium 92217) by incubation with dye for 1 hr at RT in the dark. After dye removal using Pierce Dye Removal Columns (Thermo Scientific #22858), protein was determined by Bradford assay. Labeling efficiency was determined using the dye extinction coefficient and preps were labeled with 2–3 moles of dye per mole LRRK2 for all experiments.

GFP Rab10 Q68L C-terminal His was added to supported lipid bilayers at a final concentration of 2.5 µM in STD buffer and incubated for 20 min at 37°C. After incubation, Rab-coated supported lipid bilayers were washed with STD buffer and then equilibrated with reaction buffer (20 mM HEPES pH 8, 150 mM NaCl, 5 mM MgCl$_2$, 4 mM ATP, 20 µM GTP, 20 mM creatine phosphate, 30U creatine phosphokinase). 14 nM CF633-FLAG LRRK2 was prepared in reaction buffer and allowed to equilibrate to RT for 5 min. Then, 40 s into imaging, 100 µL from the 200 uL in the reaction chamber was removed. At 60 s, 100 µL of 14 nM CF FLAG LRRK2 was added and imaged for 600 s and for 300 s for no Rab10 control.

## LRRK2 kinase activation assay

A. Rab phosphorylation. *Method #1.* Purified His-Rab8A Q67L (0.5 mg) was phosphorylated using His-MST3 kinase (0.1–0.3 mg) as described above at 30°C overnight in MST3 reaction buffer (50 mM HEPES, pH 8, 5% [v/v] glycerol, 150 mM NaCl, 5 mM MgCl$_2$, 0.2 mM TCEP, 20 µM GTP, 5 µM BSA, 0.01% Tween-20, and 2 mM ATP). Phosphorylated Rab8A (25 kDa) was then resolved from MST3 (55 kDa) by gel filtration on a 24 mL Superdex 75 10/300 column (Cytiva Life Sciences, #17517401). An additional method #2 was attempted to try to further remove trace MST3 from phosphoRab8. GST-PreScission protease was bound to glutathione agarose. His-MST3 was added to the beads and incubated overnight at 4°C. The supernatant containing free MST3 was then passed through a Nickel-NTA column to remove any uncleaved His-MST3. The pooled, untagged, MST3 supernatants were then used to phosphorylate His-Rab8A. The products of this reaction were gel filtered on Superdex 75 column as before, and phosphorylated His-Rab8A was then further purified by immobilization on nickel-NTA agarose, eluted with 500 mM imidazole after washing, and desalted as described above.

B. Kinase activation (*Chiang and Pfeffer, 2022a*; https://www.protocols.io/view/assay-for-phosphorab-activation-of-lrrk2-kinase-6qpvr4o8zgmk/v1) LRRK2 G2019S (88 nM; Thermo Fisher Scientific #A15200) or purified FLAG-LRRK2 R1441G K18A (*Adhikari et al., 2022*) was incubated with 3 µM GFP-Rab10 Q68L His or His-SUMO-Rab10 wild-type full-length substrate ±6 µM phosphorylated Rab8A Q67L in 50 mM HEPES pH 8, 5% (v/v) glycerol, 150 mM NaCl, 10 mM MgCl$_2$, 250 µM GTP, 5 µM BSA, and 2 mM ATP. No difference was detected between the two Rab10 substrates. The reaction was incubated at 30°C in a water bath. Reactions were stopped by the addition of SDS-PAGE sample buffer; MLi-2 (200 nM) was included in control reactions. Samples were analyzed by SDS-PAGE and immunoblotted for phosphoRab10. Blots were imaged using LI-COR and bands quantified using ImageJ (*Schneider et al., 2012*). The values obtained with MLi-2 were subtracted from their respective timepoints to monitor LRRK2-dependent phosphorylation; background was due to trace residual MST kinase. Values from four independent, replicate experiments were normalized to the 20 min time point and plotted together using GraphPad Prism.

## Dual Rab GTPase binding to the LRRK2 Armadillo domain

The strategy was to immobilize Rab8A, bind Armadillo domain, and then test whether Rab8A-tethered Armadillo domain could simultaneously bind phosphoRab10 (*Chiang and Pfeffer, 2022b*; https://www.protocols.io/view/assay-for-dual-rab-gtpase-binding-to-the-lrrk2-arm-81wgbypzovpk/v1). His-Rab10 Q68L 1–181 was pre-phosphorylated with His-MST3 kinase at a molar ratio of 3:1 (substrate:kinase) at 30°C for 2 hr in MST3 reaction buffer. 50 µL glutathione agarose slurry was pelleted and resuspended in 50 mM HEPES, pH 8, 5% (v/v) glycerol, 150 mM NaCl, 5 mM MgCl$_2$, 0.2 mM TCEP, 100 µM GTP, 5 µM BSA, 0.01% Tween-20 to achieve a total volume of 50 µL. GST-Rab8A Q67L (6 µM in 50 µL) was incubated with glutathione beads in reaction buffer for 30 min at RT on a rotator. The reaction was spun down at 3200 × *g* for 30 s and the supernatant discarded. His-LRRK2 Armadillo domain 1–552 in reaction buffer (or buffer alone) was added to beads to achieve a final concentration of 10 µM in 50 µL and incubated for 30 min at RT on a rotator. The reaction was spun down as before and the supernatant discarded. Phosphorylated His-Rab10 Q68L 1–181 (4 µM final) was added to beads in a final volume of 50 µL. Reactions were incubated for 30 min at RT on a rotator. The reaction was spun down at 3200 × *g* for 30 s and the supernatant discarded; reaction buffer (500 µL) was used to wash the beads twice. Proteins were eluted from the beads using 50 µL elution buffer (50 mM HEPES, pH 8, 5% [v/v] glycerol, 150 mM NaCl, 5 mM MgCl$_2$, 0.2 mM TCEP, 20 µM GTP, 50 mM reduced glutathione). The reaction was spun down at 3200 × *g* for 30 s and the supernatant was collected. Samples were then analyzed by SDS-PAGE and immunoblotted for phosphoRab10. Blots were imaged using LI-COR, and bands were quantified using ImageJ (*Schneider et al., 2012*).

## Intensity analysis of TIRF videos

Tracks of individual molecules were extracted from TIRF microscopy images using the TrackIt Fiji plugin (*Kuhn et al., 2021*) and converted to .csv files using the custom 'getTracks.m' MATLAB script (https://github.com/PfefferLab/Vides_et_al_2022; *Vides, 2022*). These files were loaded as data frames in R (*R Development Core Team, 2021*) and processed with dplyr for the binning and normalization steps. Pre-normalized intensities $I_t$ were obtained from the amplitude value fitted by TrackIt (background-corrected amplitude of the Gaussian fit of each particle). Ridge plots were produced using the ggridges package with a Gaussian Kernel density and a bandwidth of 0.2. Code used to generate each figure is available on GitHub (https://github.com/PfefferLab/Vides_et_al_2022, copy archived at swh:1:rev:2b50525ee1d48790466d35222956f16615ae96e8; *Vides, 2022*).

## Acknowledgements

This study was funded by the joint efforts of The Michael J. Fox Foundation for Parkinson's Research (MJFF) (17298 and 6986 [SRP and DRA]) and Aligning Science Across Parkinson's (ASAP) initiative. MJFF administers the grant (ASAP-000463, SRP and DRA) on behalf of ASAP and itself. Funds were also provided by the Medical Research Council (grant no. MC_UU_00018/1 [DRA]), the pharmaceutical companies supporting the Division of Signal Transduction Therapy Unit Boehringer-Ingelheim, GlaxoSmithKline, Merck KGaA (DRA), and a PhD fellowship from Consejería de Economía, Conocimiento y Empleo del Gobierno de Canarias in partnership with Fondo Social Europeo (ES-L). For the purpose of open access, the authors have applied a CC-BY public copyright license to the

Author Accepted Manuscript version arising from this submission. All primary data associated with each figure has been deposited in a repository and most can be found at https://doi.org/10.5061/dryad.3tx95x6j7; quantitation data of the blots in Figure 3--Fig. Supp. 4Figure 3—figure supplement 4 (for the bar graphs in Figures 3C and 3DFigure 3C and D) can be found at doi (10.5281/zenodo.7057419) and data for Figure 8—figure supplement 1 can be found at https://github.com/PfefferLab/Vides_et_al_2022. We are especially grateful to Dr Gheorghe Chistol for helpful discussion and Chloe Rollock for help with protein purification. We also thank the excellent technical support of the MRC protein phosphorylation and ubiquitylation unit (PPU) DNA sequencing service (coordinated by Gary Hunter), the MRC-PPU tissue culture team (coordinated by Edwin Allen), MRC-PPU Reagents and Services antibody and protein purification teams (coordinated by Dr James Hastie).

## Additional information

### Competing interests

Suzanne R Pfeffer: Senior editor, eLife. The other authors declare that no competing interests exist.

### Funding

| Funder | Grant reference number | Author |
| --- | --- | --- |
| Aligning Science Across Parkinson's Disease | ASAP-000463 | Dario R Alessi |
| Michael J Fox Foundation for Parkinson's Research | 17298 | Dario R Alessi |
| Michael J Fox Foundation for Parkinson's Research | 6986 | Dario R Alessi |
| Medical Research Council | MC_UU_00018/1 | Dario R Alessi |

The funders had no role in study design, data collection and interpretation, or the decision to submit the work for publication.

### Author contributions

Edmundo G Vides, Conceptualization, Data curation, Formal analysis, Validation, Investigation, Visualization, Writing – original draft, Writing – review and editing; Ayan Adhikari, Conceptualization, Formal analysis, Investigation, Methodology, Writing – review and editing; Claire Y Chiang, Data curation, Formal analysis, Validation, Investigation, Visualization, Methodology, Writing – original draft; Pawel Lis, Elena Purlyte, Conceptualization, Data curation, Formal analysis, Investigation, Visualization, Writing – review and editing; Charles Limouse, Data curation, Software, Formal analysis, Visualization, Writing – original draft; Justin L Shumate, Elena Spínola-Lasso, Investigation, Writing – review and editing; Herschel S Dhekne, Data curation, Formal analysis, Writing – review and editing; Dario R Alessi, Conceptualization, Resources, Data curation, Supervision, Funding acquisition, Project administration, Writing – review and editing; Suzanne R Pfeffer, Conceptualization, Data curation, Formal analysis, Supervision, Funding acquisition, Visualization, Writing – original draft, Project administration, Writing – review and editing

### Author ORCIDs

Edmundo G Vides ⬤ http://orcid.org/0000-0002-3609-9001
Ayan Adhikari ⬤ http://orcid.org/0000-0002-8525-3263
Claire Y Chiang ⬤ http://orcid.org/0000-0002-0999-9856
Pawel Lis ⬤ http://orcid.org/0000-0002-4978-7671
Justin L Shumate ⬤ http://orcid.org/0000-0002-3223-0922
Elena Spínola-Lasso ⬤ http://orcid.org/0000-0003-2207-5786
Herschel S Dhekne ⬤ http://orcid.org/0000-0002-2240-1230
Dario R Alessi ⬤ http://orcid.org/0000-0002-2140-9185
Suzanne R Pfeffer ⬤ http://orcid.org/0000-0002-6462-984X

Decision letter and Author response
Decision letter https://doi.org/10.7554/eLife.79771.sa1

## Additional files

### Supplementary files
• MDAR checklist

### Data availability
All primary data associated with each figure has been deposited in a repository; most can be found at https://doi.org/10.5061/dryad.3tx95x6j7. Quantitation data of the blots in Figure 3--figure supplement 4 (for the bar graphs in Figures 3C and 3D) can be found at doi (10.5281/zenodo.7057419). Analysis presented in Figure 8--figure supplement 1 can be found at https://doi.org/10.5281/zenodo.7108943. All code is available at https://github.com/PfefferLab/Vides_et_al_2022 (copy archived at swh:1:rev:2b50525ee1d48790466d35222956f16615ae96e8).

The following datasets were generated:

| Author(s) | Year | Dataset title | Dataset URL | Database and Identifier |
|---|---|---|---|---|
| Vides EG, Pfeffer SR | 2022 | Data from: A feed-forward pathway drives LRRK2 kinase membrane recruitment and activation | https://dx.doi.org/10.5061/dryad.3tx95x6j7 | Dryad Digital Repository, 10.5061/dryad.3tx95x6j7 |
| Limouse C, Vides EG, Adhikari A, Pfeffer SR | 2022 | PfefferLab/Vides_et_al_2022: v1.0 | https://doi.org/10.5281/zenodo.7108943 | Zenodo, 10.5281/zenodo.7108943 |
| Lis P, Alessi DR | 2022 | Figure 3–Figure Supplement 4 of the paper 'A Feed-forward Pathway Drives LRRK2 kinase Membrane Recruitment and Activation' | https://doi.org/10.5281/zenodo.7057419 | Zenodo, 10.5281/zenodo.7057419 |

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
