## [Editor Report]

This article, which is of interest to membrane biologists and colleagues in signal transduction, examines the interesting question of whether LRRK2 recruitment to membranes may regulate its activity. Membrane association involves binding to membrane-tethered Rab GTPases via LRRK2's Armadillo domain, and the authors provide an exciting and elegant feed-forward mechanism to describe how recruitment of phospho-RAB8 can promote phosphorylation of RAB10.

---

## [Decision Letter]

**Decision letter after peer review:**

Thank you for submitting your article "A Feed-forward Pathway Drives LRRK2 kinase Membrane Recruitment and Apparent Activation" for consideration by *eLife*. Your article has been reviewed by 3 peer reviewers, and the evaluation has been overseen by a Reviewing Editor and Vivek Malhotra as the Senior Editor. The reviewers have opted to remain anonymous.

Essential revisions:

1) Overall, the reviewers were positive about the overall results and were intrigued by the feed-forward model put forward. The reviewers made numerous suggestions in terms of relatively small revisions that will improve the paper (see detailed reviews). The main shortcoming is the absence of data directly addressing two important features of the feed-forward model: (1) The proposal that the increased activity of LRRK2 upon recruitment to membranes is only the result of its increased local concentration (without any contributions from a potential Rab-dependent activation); and (2) the ability of LRRK2 to simultaneously bind Rab and pRab.

2) In order to address these concerns, reviewer 2 suggests performing an experiment using the D2017A mutant together with Rab10(T73A) plus pRab10. This could both test the local concentration idea and also provide evidence, albeit indirect, of simultaneous binding to site 1 and site 2.

3) Given that reagents seem to be at hand, you should compare the kinase activity in vitro of LRRK2 WT and K17/18A, either by themselves or in the presence of pRab8/10. This would test directly whether the role played by the K17/18-pRab8/10 interaction is only a recruiting one.

*Reviewer #2 (Recommendations for the authors):*

Detailed critique:

– The authors could demonstrate the binding of Rab29 (site #1) as well as of the LRRK2 substrates Rab8a and Rab10 (sites #1 and #2) to the LRRK2 Armadillo domain. The non-LRRK2 substrate Rab7 shows no binding to any of the Rab binding sites within the LRRK2 Armadillo domain. Have the authors also tested Rab12 binding to sites #1 and #2?

Figure 1:

– The nucleotide state of the Rab proteins should be confirmed (see: major concerns).

– Since Rab7 shows no interaction (as expected) it would be good to show, that it is still correctly folded, e.g. by demonstrating nucleotide binding.

– An error estimation should be provided for the Kd determination.

Figure 2:

– The nucleotide state of the Rab proteins should be confirmed (see: major concerns).

– Hill plot provided for Rab8a in graph A: Two binding sites have been found for both, Rab8a as well as Rab10. While clear cooperativity is observed for Rab10 (panelD), the slope in panel A does not indicate a cooperative effect. It would be appreciated if the authors could comment on this and7or change the discussion/ interpretation of the data, accordingly.

– An error estimation should be provided for the Kd determination.

Figure 3:

– Panel C and D: number of interdependent data points should be provided for each condition.

– What is the rationale for preferring K439E over R399E as recommended mutation disrupting Rab29 binding?

Figure 4:

– The nucleotide state of the Rab proteins should be confirmed (see: major concerns).

– Phosphorylation after MST treatment should be demonstrated.

– An error estimation should be provided for the Kd determination.

Figure 7:

– Panel A + B: As R1441G has previously been demonstrated by the authors to show no/ reduced pS935 phosphorylation, the authors might consider using pS1292 as the more suitable marker.

– Panel E + F are not convincing: no difference in K17/18A, look both very similar.

Figure 8:

– The nucleotide state of the Rab proteins should be confirmed (see: major concerns).

– Panel C: Is there ATP present in these measurements? Was the equilibrium reached before determining the Rab10 phosphorylation levels? To ensure that the equilibrium is reached is important to avoid mixing thermodynamics with kinetic effects.

Figure 9:

– The simplified scheme is overall helpful. In contrast to the conclusions drawn in the text, it however does not consider the observation that the N-terminal Rab8a/Rab10 binding site (#2) can bind both forms, phosphorylated Rabs as well as non-phosphorylated Rabs. The authors may consider revising the figure, accordingly.

*Reviewer #3 (Recommendations for the authors):*

As mentioned in the Public Review, the main shortcoming in the manuscript is the absence of experiments testing two important components of the feed-forward model: (1) The idea that the increased activity of LRRK2 upon recruitment to membranes is only the result of its increased local concentration. The authors have not ruled out a contribution from a Rab-dependent activation of LRRK2; and (2) The ability of LRRK2 to simultaneously bind Rab and pRab. This was not tested directly, either. These two points are raised below, along with other, lesser issues that should also be addressed to strengthen the manuscript. For simplicity, the comments are organized according to the sections in the Results.

Rab29 binds to the C-terminal portion of the LRRK2 Armadillo domain

– Mg++ is not a conventional notion of valency. Please change to Mg^2+^.

– I suggest reordering the panels in Figure 1 as 1A, 1C, 1D, and 1B. Rab7, the negative control, is only mentioned in the text after panels 1A, 1C, and 1D.

Rab8A and Rab10 bind to the LRRK2 Armadillo domain

– The authors claim that their data "indicate that Rab8A binds to the same site as Rab29". The fragment they worked with is 200 amino acids long. Given that, the language should be toned down, either to say that the data suggest that the binding sites may be the same, or that the Rabs bind to the same region of the protein.

– The affinities reported for Rab8A and Rab10 in Figure 2 are surprising. Why are the differences in affinity for Rab8A between the fragment that is proposed to contain the binding site (350-550) and the one that is not (1-159) only ~2-fold? Why does Rab10 have a higher affinity for the full fragment (1-552) than for the 350-550 fragment that is supposed to contain the binding site? This was not the case with Rab29, where the difference between the two non-overlapping fragments was ~20-fold (Figure 1). Are the binding sites different for the different Rabs? The authors should discuss this and propose some explanations.

Residues critical for Rab GTPase binding to LRRK2 residues 350-550, Site #1

– Figure 3: Please mark the Switch II region on Rab in your AF model. This will be helpful for those readers less familiar with Rabs.

– 'Intimate' is not a technical way of referring to contacts. The authors should describe how contacts are measured and then define "close" contacts as those below whatever distance threshold they choose.

– Figure 3D is missing a description in the figure legend. The legend is also missing an explanation of what the red and grey boxes represent in panels C and D.

– Figure 3C, D: Please explain the red asterisk in the figure legend. They are useful information but I was confused by them until I reached the explanation in the main text.

– Figure 3-Figure Supp.1: I would suggest making the grey bars representing the different constructs lighter to increase the contrast with the green ones.

– In Figure 3-Figure Supp4 (page 41): The contrast of the bottom row in each gel ["HA (Rab29)"] is much higher than in the others. What is the reason for this difference?

– Do these residues affect binding to Rab8A and/or 10?

PhosphoRab binding to LRRK2, Site #2

– The authors should provide some indication of the extent to which Rabs were phosphorylated by MST3.

PhosphoRab-LRRK2 interaction increases rates of kinase recovery

– How was the re-phosphorylation quantified? Based on the gels it looks as though all variants did a similarly good job at re-phosphorylating Rab10. It would be important to quantify the levels of LRRK2 throughout the experiment as well. The bands at 120 min for both LRRK2 R1441G K17/18A and LRRK2 appear weaker; the authors need to show that changes in the recovery from MLi-2 cannot simply be explained by differences in LRRK2 levels.

– An important component of the feed-forward mechanism proposed is the increase in local LRRK2 concentration brought about by the higher affinity between phosphorylated Rab8/10 and Site #2. While the model implies that the only role played by this interaction is to recruit LRRK2, this was not tested directly. Neither was the effect of the K17/18A mutations tested on LRRK2's activity. Given that reagents seem to be at hand, the authors should compare the kinase activity in vitro of LRRK2 WT and K17/18A, either by themselves or in the presence of pRab8/10. This would test directly whether the role played by the K17/18-pRab8/10 interaction is only a recruiting one.

Cooperative LRRK2 membrane recruitment on Rab-decorated planar lipid bilayers

– The wording "Binding of phosphoRab reaction products…" is a bit confusing. Presumably, phosphoRab is the reaction product of the phosphorylation reaction, but there is no "phosphoRab reaction".

– There is a reference to Figure 3B ("This indicates that at least Rab10…") that seems incorrect.

– The significant reduction in membrane binding by LRRK2(D2017A) relative to R1441G and K17A/K18A/R1441G is an interesting observation, but I am not sure the interpretation suggested by the authors--that binding Site #1 may be more accessible in an active LRRK2 conformation--is the only, or even more likely one. I am assuming that the experimental setup makes it possible for LRRK2 to phosphorylate the membrane-bound Rab10, thus triggering the feed-forward loop. (If that is not the case, it would be useful to explain why in the main text so other readers do not make the same assumption.) In my assumption is correct, LRRK2(D2017A) could have two different (and co-existing) effects: (1) It would prevent the formation of pRab10 and thus the higher affinity pRab10-Site #2 interaction; and (2) It will affect LRRK2's conformation. It seems to me that the proposed model could be strengthened by a few additional experiments. These would involve repeating the membrane recruitment assay with Rab10 variants. Using pRab10 would bypass LRRK2(D2017A)'s inability to generate pRab10 and thus test whether LRRK2's conformation plays a role in recruitment via Site #2. An even more informative experiment would involve using a combination of pRab10 and Rab10(T73A) as this would both bypass D2017A's lack of activity, but it would also allow for engagement of both Site #1 and Site #2 without allowing for an increased in the density of pRab10 on the membrane. Comparing the results of this experiment with one using only pRab10 (but with the same total Rab10 density on the membrane) could also provide evidence for LRRK2's ability to bind two Rab's (Rab and pRab) at the same time (one of the issues raised in the Public Review and at the beginning of these comments).

---

## [Author Response]

Essential revisions:1) Overall, the reviewers were positive about the overall results and were intrigued by the feed-forward model put forward. The reviewers made numerous suggestions in terms of relatively small revisions that will improve the paper (see detailed reviews). The main shortcoming is the absence of data directly addressing two important features of the feed-forward model: (1) The proposal that the increased activity of LRRK2 upon recruitment to membranes is only the result of its increased local concentration (without any contributions from a potential Rab-dependent activation); and (2) the ability of LRRK2 to simultaneously bind Rab and pRab.2) In order to address these concerns, reviewer 2 suggests performing an experiment using the D2017A mutant together with Rab10(T73A) plus pRab10. This could both test the local concentration idea and also provide evidence, albeit indirect, of simultaneous binding to site 1 and site 2.

Unfortunately, we cannot use T73A Rab10 as in our previous work (Dhekne et al., *eLife* 2018) we showed that this mutant is not recognized by Rab prenyl transferase in cells (thus may not be able to bind nucleotide) and it cannot rescue a cilia phenotype in Rab10 knockout cells. Lack of cilia phenotype rescue holds true for the equivalent Rab8 mutant. Nevertheless, we used another strategy to show that LRRK2 Arm domain can simultaneously bind to pRab and non-phosphoRab at Sites 1 and 2 (see below): GST-immobilized Rab8A can bind Arm domain and pRab10 can bind this Arm domain.

3) Given that reagents seem to be at hand, you should compare the kinase activity in vitro of LRRK2 WT and K17/18A, either by themselves or in the presence of pRab8/10. This would test directly whether the role played by the K17/18-pRab8/10 interaction is only a recruiting one.

These are hard experiments as each LRRK2 prep is of low yield and LRRK2 is a weak kinase. The good news is that thanks to the reviewer comments, we have been able to show that phosphorylated Rab8 stimulates the in vitro kinase activity of LRRK2 on Rab10 by 4 fold, and this requires the presence of Lysine 18. Thus, we have changed the title and text to remove, “apparent” activation as we can now state that it is bona fide enzymatic activation. We thank the reviewer for this suggestion which adds great impact to the overall story.

Reviewer #2 (Recommendations for the authors):Detailed critique:– The authors could demonstrate the binding of Rab29 (site #1) as well as of the LRRK2 substrates Rab8a and Rab10 (sites #1 and #2) to the LRRK2 Armadillo domain. The non-LRRK2 substrate Rab7 shows no binding to any of the Rab binding sites within the LRRK2 Armadillo domain. Have the authors also tested Rab12 binding to sites #1 and #2?

We have not tested the binding of Rab12 to these sites.

Figure 1:– The nucleotide state of the Rab proteins should be confirmed (see: major concerns).– Since Rab7 shows no interaction (as expected) it would be good to show, that it is still correctly folded, e.g. by demonstrating nucleotide binding.

Rab7 is fully folded as determined by Tm determination (65°C).

– An error estimation should be provided for the Kd determination.

We have included error estimations for the Kd determinations as requested in a summary Table 1.

Figure 2:– The nucleotide state of the Rab proteins should be confirmed (see: major concerns).– Hill plot provided for Rab8a in graph A: Two binding sites have been found for both, Rab8a as well as Rab10. While clear cooperativity is observed for Rab10 (panelD), the slope in panel A does not indicate a cooperative effect. It would be appreciated if the authors could comment on this and7or change the discussion/ interpretation of the data, accordingly.

As described in the text, the interaction of non-phosphoRabs with 1-159 is likely not possible in a longer fragment as those Rabs seem to bind right at that fragments’s absolute C-terminus that would not be available in a longer fragment (predicted by Alphafold). The text has been clarified accordingly.

Included in Table 1.

Figure 3:– Panel C and D: number of interdependent data points should be provided for each condition.

We corrected the legend as requested–these are from the western blots shown in Figure 3 Supp 4.

– What is the rationale for preferring K439E over R399E as recommended mutation disrupting Rab29 binding?

The R1441G data look better for K439E but yes, R399E looks better for wild type. The text has been clarified.

Figure 4:– The nucleotide state of the Rab proteins should be confirmed (see: major concerns).– Phosphorylation after MST treatment should be demonstrated.

We have demonstrated this to be >90% by Phostag gels in Figure 6, Dhekne et al., 2021 Life Science Alliance Mar 2021, 4 (5) e202101050; DOI: 10.26508/lsa.202101050

– An error estimation should be provided for the Kd determination.

Provided as requested Table 1.

Figure 7:

– Panel A + B: As R1441G has previously been demonstrated by the authors to show no/ reduced pS935 phosphorylation, the authors might consider using pS1292 as the more suitable marker.

Actually the pS935 does not add anything here and so we removed it from the figure

– Panel E + F are not convincing: no difference in K17/18A, look both very similar.

We do not expect a big difference between E and F–what is different is the recovery rates between the blue lines and orange lines in each figure section. We see a bigger difference for the R1441G and include wild type LRRK2 to be complete.

Figure 8:– The nucleotide state of the Rab proteins should be confirmed (see: major concerns).– Panel C: Is there ATP present in these measurements? Was the equilibrium reached before determining the Rab10 phosphorylation levels? To ensure that the equilibrium is reached is important to avoid mixing thermodynamics with kinetic effects.

4mM ATP is present throughout, together with an ATP regenerating system. No phosphorylation is measured–only activity-dependent, LRRK2-bilayer association. Each TIRF microscopy video displays 10min binding, and equilibrium was reached for all conditions in terms of LRRK2 binding to the Rabdecorated bilayer.

Figure 9:– The simplified scheme is overall helpful. In contrast to the conclusions drawn in the text, it however does not consider the observation that the N-terminal Rab8a/Rab10 binding site (#2) can bind both forms, phosphorylated Rabs as well as non-phosphorylated Rabs. The authors may consider revising the figure, accordingly.

See above–we do not think non-phospho Rabs bind site #2 physiologically.

Reviewer #3 (Recommendations for the authors):As mentioned in the Public Review, the main shortcoming in the manuscript is the absence of experiments testing two important components of the feed-forward model: (1) The idea that the increased activity of LRRK2 upon recruitment to membranes is only the result of its increased local concentration. The authors have not ruled out a contribution from a Rab-dependent activation of LRRK2; and (2) The ability of LRRK2 to simultaneously bind Rab and pRab. This was not tested directly, either. These two points are raised below, along with other, lesser issues that should also be addressed to strengthen the manuscript. For simplicity, the comments are organized according to the sections in the Results.Rab29 binds to the C-terminal portion of the LRRK2 Armadillo domain– Mg++ is not a conventional notion of valency. Please change to Mg^2+^.

Changed as requested

– I suggest reordering the panels in Figure 1 as 1A, 1C, 1D, and 1B. Rab7, the negative control, is only mentioned in the text after panels 1A, 1C, and 1D.

We reordered the text so that the panels will be described in order.

Rab8A and Rab10 bind to the LRRK2 Armadillo domain– The authors claim that their data "indicate that Rab8A binds to the same site as Rab29". The fragment they worked with is 200 amino acids long. Given that, the language should be toned down, either to say that the data suggest that the binding sites may be the same, or that the Rabs bind to the same region of the protein.

The reviewer is correct and we have toned down the language and sought any other sites with Alphafold and find none (shown in supplement).

– The affinities reported for Rab8A and Rab10 in Figure 2 are surprising. Why are the differences in affinity for Rab8A between the fragment that is proposed to contain the binding site (350-550) and the one that is not (1-159) only ~2-fold?

This is because there are multiple sites as now more clearly described in the text.

Why does Rab10 have a higher affinity for the full fragment (1-552) than for the 350-550 fragment that is supposed to contain the binding site?

This suggests the possibility that there is more than one site in the 1-552 fragment which could include low affinity for the N-terminus.

This was not the case with Rab29, where the difference between the two non-overlapping fragments was ~20-fold (Figure 1). Are the binding sites different for the different Rabs?

Yes, we suspect that exact binding sites are different for the different Rabs.

The authors should discuss this and propose some explanations.

More discussion has been added.

Residues critical for Rab GTPase binding to LRRK2 residues 350-550, Site #1– Figure 3: Please mark the Switch II region on Rab in your AF model. This will be helpful for those readers less familiar with Rabs.

Now colored Orange (SW-I) and Yellow (SW-II) as requested

– 'Intimate' is not a technical way of referring to contacts. The authors should describe how contacts are measured and then define "close" contacts as those below whatever distance threshold they choose.

Corrected thanks

– Figure 3D is missing a description in the figure legend. The legend is also missing an explanation of what the red and grey boxes represent in panels C and D.

Corrected thanks

– Figure 3C, D: Please explain the red asterisk in the figure legend. They are useful information but I was confused by them until I reached the explanation in the main text.

Corrected thanks

– Figure 3-Figure Supp.1: I would suggest making the grey bars representing the different constructs lighter to increase the contrast with the green ones.

Corrected thanks

– In Figure 3-Figure Supp4 (page 41): The contrast of the bottom row in each gel ["HA (Rab29)"] is much higher than in the others. What is the reason for this difference?

The reason for that is that we probed each membrane with pRab10 (rabbit), Rab10 (mouse) and HA (rat) antibodies. pRab10 and Rab10 signals were developed using 800 (anti-rabbit) and 680 (anti-mouse) channels in LICOR, whereas the HA (showing Rab29 expression) was developed using ECL (anti-rat), which gives higher contrast.

– Do these residues affect binding to Rab8A and/or 10?

Likely yes per Alphafold but not yet tested as carefully

PhosphoRab binding to LRRK2, Site #2– The authors should provide some indication of the extent to which Rabs were phosphorylated by MST3.

See response to reviewer 2 above–90% by Phostag gel (Dhekne et al., 2021)

PhosphoRab-LRRK2 interaction increases rates of kinase recovery– How was the re-phosphorylation quantified? Based on the gels it looks as though all variants did a similarly good job at re-phosphorylating Rab10. It would be important to quantify the levels of LRRK2 throughout the experiment as well. The bands at 120 min for both LRRK2 R1441G K17/18A and LRRK2 appear weaker; the authors need to show that changes in the recovery from MLi-2 cannot simply be explained by differences in LRRK2 levels.

We did not further normalize to LRRK2 levels but normalization actually improves the argument.

– An important component of the feed-forward mechanism proposed is the increase in local LRRK2 concentration brought about by the higher affinity between phosphorylated Rab8/10 and Site #2. While the model implies that the only role played by this interaction is to recruit LRRK2, this was not tested directly. Neither was the effect of the K17/18A mutations tested on LRRK2's activity. Given that reagents seem to be at hand, the authors should compare the kinase activity in vitro of LRRK2 WT and K17/18A, either by themselves or in the presence of pRab8/10. This would test directly whether the role played by the K17/18-pRab8/10 interaction is only a recruiting one.

We thank the referee for this suggestion and can now state that pRab8A actually enhances the rate of Rab10 phosphorylation by LRRK2. Also, K18A LRRK2 is of similar activity as LRRK2 in the new figure 9.

Cooperative LRRK2 membrane recruitment on Rab-decorated planar lipid bilayers– The wording "Binding of phosphoRab reaction products…" is a bit confusing. Presumably, phosphoRab is the reaction product of the phosphorylation reaction, but there is no "phosphoRab reaction".

Corrected thanks

– There is a reference to Figure 3B ("This indicates that at least Rab10…") that seems incorrect.

Corrected thanks

– The significant reduction in membrane binding by LRRK2(D2017A) relative to R1441G and K17A/K18A/R1441G is an interesting observation, but I am not sure the interpretation suggested by the authors--that binding Site #1 may be more accessible in an active LRRK2 conformation--is the only, or even more likely one. I am assuming that the experimental setup makes it possible for LRRK2 to phosphorylate the membrane-bound Rab10, thus triggering the feed-forward loop. (If that is not the case, it would be useful to explain why in the main text so other readers do not make the same assumption.)

Correct.

In my assumption is correct, LRRK2(D2017A) could have two different (and co-existing) effects: (1) It would prevent the formation of pRab10 and thus the higher affinity pRab10-Site #2 interaction; and (2) It will affect LRRK2's conformation. It seems to me that the proposed model could be strengthened by a few additional experiments. These would involve repeating the membrane recruitment assay with Rab10 variants. Using pRab10 would bypass LRRK2(D2017A)'s inability to generate pRab10 and thus test whether LRRK2's conformation plays a role in recruitment via Site #2. An even more informative experiment would involve using a combination of pRab10 and Rab10(T73A) as this would both bypass D2017A's lack of activity, but it would also allow for engagement of both Site #1 and Site #2 without allowing for an increased in the density of pRab10 on the membrane. Comparing the results of this experiment with one using only pRab10 (but with the same total Rab10 density on the membrane) could also provide evidence for LRRK2's ability to bind two Rab's (Rab and pRab) at the same time (one of the issues raised in the Public Review and at the beginning of these comments).

The non-phosphorylatable Rab8A and Rab10 mutants fail to reverse their respective ciliation phenotypes observed in Rab knockout cells and are poor substrates for Rab geranylgeranyl transferase. Moreover, unlike the native phosphorylated Rab8A or Rab10 proteins, Rab8A-TE and Rab10-TE fail to show enhanced interaction with their RILPL1 effector. Thus, TA and TE mutants do not reflect the desired states of Rab8A or Rab10 and cannot be used. Other experiments have been added to demonstrate that two Rabs can simultaneously bind to Site #1 and Site #2. Future work will continue to investigate the molecular events detected.

In summary, thanks to the reviewers for really thoughtful and helpful comments!